## RESEARCH ARTICLE

# Ribosome-binding protein 1 maintains peroxisome biogenesis

Kaneez Fatima[1], Helena Vihinen[2], Ani Akpinar[1], Tamara Somborac[1], Anja Paatero[1], Eija Jokitalo[2], Ville Paavilainen[1], Pekka Katajisto[3] and Svetlana Konovalova[1,*]

## ABSTRACT

Peroxisomes are single-membrane-bound organelles essential for human health, yet the mechanisms of peroxisome biogenesis are not fully understood. Here using a systematic double screening approach, we identified ribosome-binding protein 1 (RRBP1) as a novel peroxisome biogenesis factor in human cells. Deletion of RRBP1 in HEK293T cells led to a reduction in both peroxisome number and peroxisomal protein levels as well as in defects in processing of peroxisomal matrix proteins, such as ACOX1 and thiolase. However, cell proliferation and protein translation were not altered in cells lacking RRBP1. RRBP1 depletion did not affect peroxisome–endoplasmic reticulum (ER) contact sites, and pexophagy did not contribute to the reduction of peroxisomes in RRBP1 knockout cells. Instead, in the absence of RRBP1, peroxisomal proteins were processed by proteasomal degradation, suggesting that RRBP1 plays a role in the insertion of these proteins into ER membranes and their stabilization. Altogether, our results show that RRBP1 promotes peroxisome biogenesis in human cells, highlighting the power of systematic approaches in discovering novel factors of organellar biogenesis.

KEY WORDS: Peroxisome, RRBP1, CRISPR/Cas9, Peroxisome biogenesis

## INTRODUCTION

Peroxisomes are single-membrane-bound organelles, essential for lipid breakdown and β-oxidation of very long-chain fatty acids. Peroxisomes can originate through two distinct mechanisms: *de novo* biogenesis and fission. *De novo* peroxisome biogenesis begins with the assembly of peroxisomal membrane receptors PEX3 and PEX16 at the endoplasmic reticulum (ER) and mitochondria, creating pre-peroxisomal vesicles that fuse. Following fusion, the newly formed peroxisome continues to import peroxisomal membrane proteins via the PEX19-mediated pathway. Once the import machinery is established on the membrane, peroxisomes can import matrix proteins, which are translated on free ribosomes and transported post-translationally from the cytosol. This process leads to the maturation of pre-peroxisomes into fully functional peroxisomes (Sugiura et al., 2017; Kim et al., 2006; Farré et al., 2019;

[1]Institute of Biotechnology, Helsinki Institute of Life Science (HiLIFE), University of Helsinki, 00014 Helsinki, Finland. [2]Electron Microscopy Unit, Institute of Biotechnology, Helsinki Institute of Life Science (HiLIFE), University of Helsinki, 00014 Helsinki, Finland. [3]Faculty of Biological and Environmental Sciences, University of Helsinki, 00014 Helsinki, Finland.

*Author for correspondence (svetakonovalova@gmail.com)

 S.K., 0000-0001-7363-0271

Goldman and Blobel, 1978; Hasan et al., 2013; Rachubinski et al., 1984).

Most of the peroxisomal matrix proteins contain PTS1 or PTS2 targeting sequences, which are recognized by the cytosolic peroxins PEX5 and PEX7, respectively (Walter and Erdmann, 2019; Okumoto et al., 2018; Farré et al., 2019). The translocation of substrate proteins into the peroxisomal matrix is mediated by factors localized on the peroxisomal membrane such as PEX14 and PEX13 (Schliebs et al., 1999; Albertini et al., 1997; Distel et al., 1996; Matsumoto et al., 2003). Although studies in yeast (Akşit and van der Klei, 2018; Erdmann et al., 1989) and mammalian cells (Fang et al., 2004; Fujiki et al., 2014) have identified core components required for peroxisomal matrix import, the discovery of additional factors in recent studies (Baldwin et al., 2021; Guillén-Samander et al., 2021) suggests that more remain to be identified.

To systematically identify factors involved in biogenesis of peroxisomes in mammalian cells we first utilized APEX2 proximity labeling followed by mass spectrometry as a powerful tool to study protein–protein interactions (Lam et al., 2015). Using this approach, we identified factors that specifically interact with peroxisomal matrix cargo. Secondly, we tested these factors by CRISPR/Cas9 knockout (KO) microscopy screening. Using this double screening approach we revealed ribosome-binding protein 1 (RRBP1, also known as p180/ribosome receptor), as a novel peroxisome biogenesis factor.

Using KO and knock-down models, we demonstrated that RRBP1 depletion results in peroxisome loss. Notably, RRBP1 KO did not affect cell proliferation or protein translation, and we confirmed that pexophagy was not responsible for peroxisome loss in these cells. Thus, we established that RRBP1 is essential for maintaining peroxisome number in human cells. Although RRBP1 is an ER membrane protein, its depletion did not significantly alter the abundance of the contacts between ER and peroxisomes. However, the inhibition of the proteolytic activity with the proteasome inhibitor MG132 revealed that peroxisomal proteins undergo enhanced proteasomal degradation in RRBP1 KO cells. These findings suggest that RRBP1 is involved in early induction of peroxisome biogenesis by regulating the initial membrane insertion of key peroxisomal components into the ER membrane.

## RESULTS

### Systematic identification of peroxisome biogenesis factors using a double screening approach

To systematically identify factors required for protein import into the peroxisomal matrix, we first used a proximity-labeling approach followed by mass spectrometry (Lam et al., 2015) (Fig. 1A). Ascorbate peroxidase (APEX2) was fused to ePTS1 (DeLoache et al., 2016), the enhanced peroxisomal targeting signal type 1 recognized by PEX5, directing the construct to the peroxisomal matrix. The GFP–APEX2–ePTS1 construct was stably expressed in HEK293T control cells or cells lacking peroxisomes (PEX3 KO or PEX19 KO cells; Fig. S1A). As expected, the GFP–APEX2–ePTS1 construct was recognized by the peroxisomal protein import

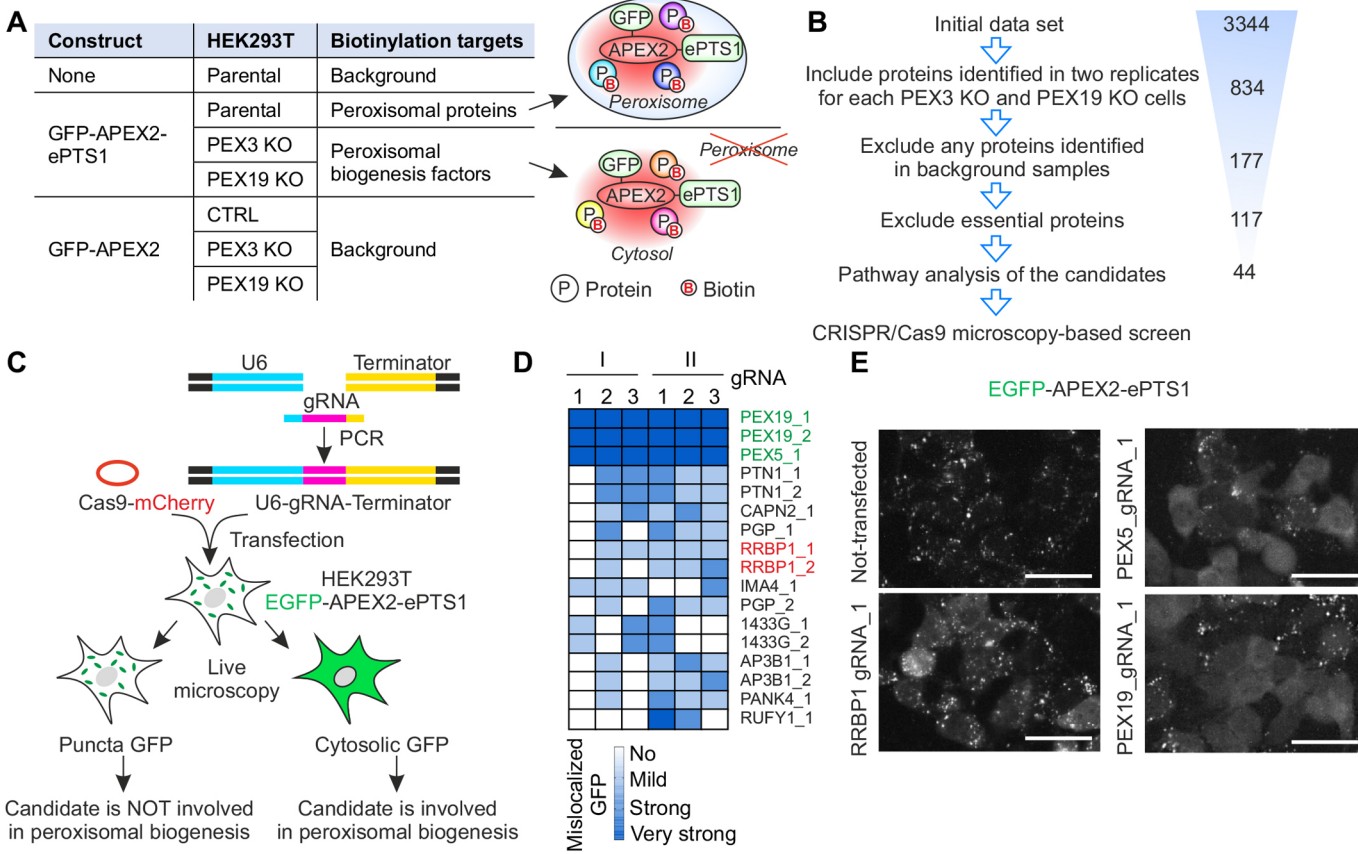

**Fig. 1. Systematic identification of peroxisome biogenesis factors using proximity labeling and CRISPR/Cas9 microscopy-based screen.**
(A) Overview of the APEX2-based proximity labeling approach to identify peroxisome biogenesis factors. B, biotin; P, protein. (B) Steps in filtering of the mass spectrometry data to identify peroxisome biogenesis factors with high confidence. Number of candidates at each step is indicated. (C) Overview of the CRISPR/Cas9 microscopy-based screen. (D) Heat map of CRISPR/Cas9 microscopy assessing the phenotype of HEK293T cells. HEK293T cells stably expressing GFP–APEX2–ePTS1 were transiently transfected with gRNAs and Cas9–mCherry plasmid as shown in C. Two rounds of transfection were performed (I and II). Each round the phenotype of the cells was assessed three times at 7, 10 and 14 days after transfection using live-cell fluorescence microscopy. Cells were manually classified based on GFP localization into 'no' (GFP localized exclusively to peroxisomes, no mislocalization), 'mild' (some mislocalized GFP), 'strong' (substantial mislocalized GFP) and 'very strong' (extensive mislocalized GFP) categories. gRNAs targeting PEX5 or PEX19 were used as positive controls (in green). The full list of the screened gRNAs is in Fig. S1D. (E) Live-cell fluorescence microscopy images of HEK293T cells stably expressing GFP–APEX2–ePTS1 before (not-transfected) and after 13 days of transfection with Cas9–mCherry plasmid and gRNA targeting RRBP1, PEX5 or PEX19. The localization of peroxisomal targeted construct GFP–APEX2–ePTS1 was visualized using the GFP channel. Scale bars: 50 μm. Images are representative of five experiments.

machinery and delivered to the peroxisomal matrix in control cells. In contrast, in PEX3 KO or PEX19 KO cells, which lack peroxisomes, the construct remained in the cytosol (Fig. S1B). Under these conditions, factors involved in recognizing and trafficking ePTS1-containing proteins would be expected to accumulate near the GFP–APEX2–ePTS1 construct. Therefore, proteins specifically enriched in proximity to GFP–APEX2–ePTS1 in PEX3 KO or PEX19 KO cells represent strong candidates for peroxisome biogenesis factors. To exclude non-specifically labeled proteins we used parental cells not expressing GFP–APEX2–ePTS1 and cells stably expressing non-targeted GFP–APEX2 as negative control samples.

Biotinylated proteins were extracted from cell lysates using streptavidin beads and analyzed by mass spectrometry. Proteins identified in both of the two replicates performed for each of the two KO cell lines, PEX3 KO and PEX19 KO, were considered as significant hits. To identify highly specific interactors of GFP–APEX2–ePTS1 in cells lacking peroxisomes, we excluded any protein identified in the negative control samples (Table S1). Next, we excluded essential proteins (Table S2). Following pathway analysis of

the remaining candidates, we identified 44 proteins that are potentially involved in peroxisome biogenesis (Fig. 1B; Fig. S1C; Table S3). Surprisingly, we did not detect PEX5 among the identified proteins, despite its role as the key factor recognizing PTS1. Notably, PEX5 was not identified as a specific interactor of GFP–APEX2–ePTS1 even in parental cells, suggesting that a combination of technical limitations in the labeling reaction and the unique biological properties of PEX5 may have prevented its detection in the APEX2 proximity labeling assay.

To ensure that only factors with a functional role in peroxisome biogenesis were identified, we performed a secondary CRISPR/Cas9-based microscopy screen of the candidates from the proximity labeling assay (Fig. 1C). For each candidate, two guide RNAs (gRNAs) were designed and co-transfected with Cas9 into HEK293T cells stably expressing GFP–APEX2–ePTS1, which served as a reporter cell line for peroxisome biogenesis. Live-cell fluorescence microscopy was performed to examine the localization of GFP in the transfected cells, in which redistribution of GFP from peroxisomes to the cytosol indicated disruption of peroxisome biogenesis (Fig. 1C). The phenotype of the cells was analyzed using live-cell microscopy, and

the cells were manually classified into four categories based on GFP localization: 'no' (GFP localized exclusively to peroxisomes, no mislocalization), 'mild' (some mislocalized GFP), 'strong' (substantial mislocalized GFP) and 'very strong' (extensive mislocalized GFP). Each gRNA was tested in two independent rounds of transfection, with analyses performed at days 7, 10 and 14 after transfection. As positive controls, gRNAs targeting known peroxisome assembly factors, such as PEX5 and PEX19, produced robust mislocalization of GFP–APEX2–ePTS1 (Fig. 1D,E; Fig. S1D) without altering its overall expression level (Fig. S1E,F). Among the hits, KO of ribosome-binding protein 1 (RRBP1) caused strong mislocalization of GFP–APEX2–ePTS1 into the cytoplasm (Fig. 1D,E), suggesting its role in peroxisomal biogenesis.

RRBP1, also known as p180/ribosome receptor, is a transmembrane ER protein. RRBP1 was originally identified as a ribosome receptor for the ER (Savitz and Meyer, 1990) and has also been proposed to regulate local translation in axons (Koppers et al., 2024). Recent studies suggest that RRBP1 plays a role in tethering ER with mitochondria (Anastasia et al., 2021; Cardoen et al., 2024; Hung et al., 2017) and regulates mitophagy (Killackey et al., 2023). However, RRBP1 has not been linked to peroxisomal function previously. Therefore, RRBP1 emerged as a particularly promising candidate due to its ER localization and established role in ribosome binding and membrane protein translocation – processes closely tied to organelle biogenesis. Together with the strong phenotype observed in our screen (Fig. 1D,E), these features provided the rationale for selecting RRBP1 for detailed functional characterization as a potential peroxisome biogenesis factor.

### RRBP1 is required for the maintenance of peroxisome number in human cells

To investigate the involvement of RRBP1 in the maintenance of peroxisomes we generated a clonal RRBP1 KO HEK293T cell line expressing the GFP–APEX2–ePTS1 construct by CRISPR/Cas9 gene editing. To achieve a complete KO of the RRBP1 gene we used a single-guide RNA (sgRNA) targeting exon 2, which is common to all isoforms of RRBP1. We confirmed the complete depletion of the RRBP1 protein in the generated RRBP1 KO cell line using immunoblotting (Fig. 2A). We analyzed RRBP1 KO HEK293T cells expressing GFP–APEX2–ePTS1 and its parental cell line as a control. In addition, as positive controls we used cells lacking essential peroxisome biogenesis factors, PEX3 or PEX19, generated in our previous study (Somborac et al., 2023). We observed that the RRBP1 KO cells have significantly lower steady-state protein levels of the peroxisomal matrix protein catalase, as well as the peroxisomal membrane proteins PEX3, PEX19, PEX14 and PXMP2, compared to the parental cell line (Fig. 2A,B; Fig. S3D). In contrast, the protein level of PEX5, an essential peroxisomal matrix protein factor, was increased upon RRBP1 KO (Fig. 2A,B).

Next, we analyzed peroxisomal function in RRBP1 KO cells by assessing peroxisomal matrix protein import. Although most of the peroxisomal proteins are translated in the cytosol in their mature form, some undergo processing upon import to peroxisomes. Once imported to the peroxisomal matrix, the 72 kDa cytosolic unprocessed acyl-CoA oxidase 1 (ACOX1), a PTS1-targeted peroxisomal matrix protein, is cleaved into its 50 kDa and 22 kDa mature forms (Kurochkin et al., 2007). Similarly, the 44 kDa unprocessed 3-ketoacyl-CoA thiolase (ACAA1), a PTS2-targeted peroxisomal matrix protein, is processed to 42 kDa upon impot to peroxisomes (Miura et al., 1984). Using immunoblotting, we showed that the processing of both ACOX1 and ACAA1 was largely abolished in PEX3 KO cells and PEX19 KO cells, which

lack peroxisomes (Fig. 2A–C), confirming that this assay can be efficiently used to analyze peroxisomal function, as shown previously (Klouwer et al., 2021; Kurochkin et al., 2007). Interestingly, a significant decrease in the processing of ACOX1 and ACAA1 was observed in the RRBP1 KO cells (Fig. 2A–C).

Notably, the PEX3 KO cells and PEX19 KO cells had significantly lower RRBP1 protein levels compared to their parental cells, suggesting that RRBP1 protein is responsive to the absence of peroxisomes. We also observed a slight decrease in the protein levels of the ER proteins calnexin and VAPB in RRBP1 KO cells compared to parental cells (Fig. 2D), suggesting that RRBP1 KO might have a mild impact on the ER. We observed no effect of RRBP1 KO on the steady-state level of the mitochondrial protein TOM20 (also known as TOMM20; Fig. 2E), suggesting that mitochondrial mass is not affected in cells lacking RRBP1.

To validate our findings on RRBP1 KO cells we performed transient knock down of RRBP1 using endoribonuclease-prepared short interfering RNA (esiRNA) in HEK293T cells stably expressing GFP–APEX2–ePTS1. GFP esiRNA was used as a negative control, whereas esiRNA targeting a peroxisomal import factor essential for the assembly of functional peroxisomes, PEX5, was used as a positive control. In cells silenced for RRBP1, protein levels of ACAA1 and catalase were reduced (Fig. 3A,B,D) and the processing of ACOX1 was significantly decreased, similar to the effect seen upon PEX5 silencing (Fig. 3C). Importantly, similar to RRBP1 KO cells, we showed mislocalization of GFP–APEX2–ePTS1 construct to the cytosol in cells with acute silencing of RRBP1 (Fig. 3E). Thus, we demonstrated that loss of RRBP1 leads to peroxisomal deficiency, indicating that RRBP1 plays a specific role in peroxisome maintenance.

### RRBP1 KO causes a decreased number of peroxisomes in human cells

Next, we investigated whether peroxisome number is affected by the loss of RRBP1. We used RRBP1 KO HEK293T cells stably expressing GFP–APEX2–ePTS1 and its parental cell line as a negative control. Live-cell fluorescence microscopy confirmed the absence of RRBP1 in KO cells and revealed a reduced number of GFP-positive puncta representing mature peroxisomes compared to the parental cells (Fig. S1G). In addition, GFP–APEX2–ePTS1 was partially localized to cytosol in cells lacking RRBP1, whereas in parental HEK293T cells the construct was correctly localized to peroxisomes, confirming our result from the CRISPR/Cas9 screen (Fig. 1D). These data collectively show that the import of peroxisomal matrix proteins is compromised in RRBP1 KO cells.

To study whether RRBP1 KO also affects peroxisome number, we performed high-resolution immunofluorescence confocal microscopy of RRBP1 KO cells using PMP70 (also known as ABCD3) and PEX14 as a membrane markers and catalase as a matrix marker of peroxisomal structures. As a positive control we used PEX3 KO or PEX19 KO HEK293T cells, which lack peroxisomes. The numbers of PMP70-, PEX14- or catalase-positive structures per cell were reduced in RRBP1 KO cells compared to the parental cells (Fig. 4A–E), confirming that RRBP1 is required for maintaining peroxisome number. Notably, cell volume was unaffected by RRBP1 KO, excluding this as a factor contributing to the reduced peroxisome abundance (Fig. 4F). Altogether these results show that RRBP1 is an important factor involved in the maintenance of peroxisome number in human cells.

### RRBP1 does not localize to peroxisomes
Our data indicate that RRBP1 maintains peroxisome biogenesis. Previously, RRBP1 has been described as an ER transmembrane

Journal of Cell Science

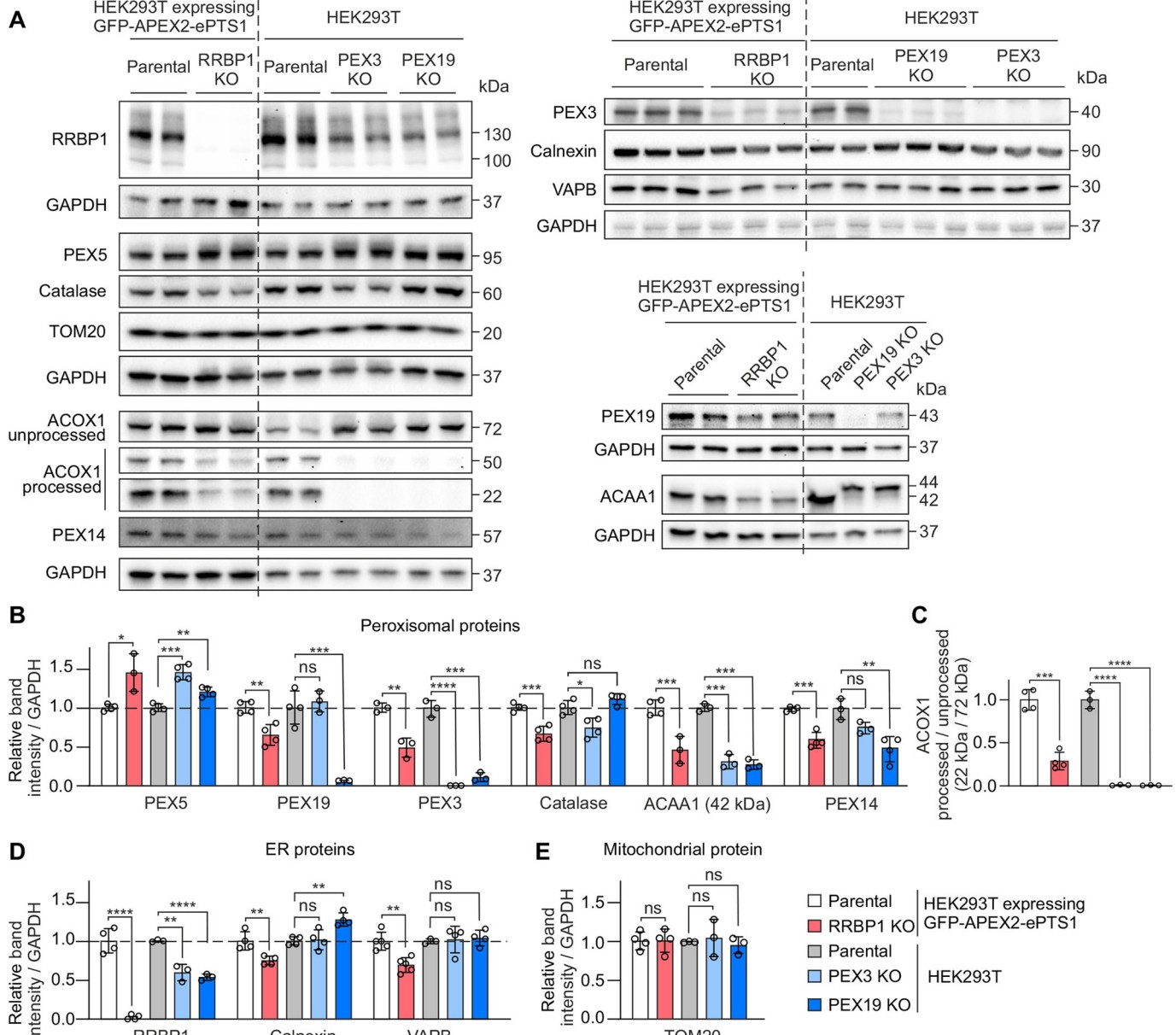

**Fig. 2. RRBP1 is required for normal peroxisome maintenance in human cells.** (A) Western blot analysis of the indicated proteins in RRBP1 KO HEK293T cells stably expressing GFP–APEX2–ePTS1. Parental, PEX3 and PEX19 KO HEK293T cells not expressing the peroxisomal-targeted construct were used as controls. GAPDH was used as a loading control. (B) Quantification of peroxisomal protein levels using western blot images as represented in A ($n$=3–4). (C) Quantification of ACOX1 processing using western blot images as represented in A ($n$=3–4). (D) Quantification of ER protein levels using western blot images as represented in A ($n$=3–5). (E) Quantification of TOM20 protein level using western blot images as represented in A ($n$=3–4). Each western blot analysis was performed at least three times. Values are normalized to the corresponding parental cell line, set to 1. In all graphs, data are presented as mean±s.d. *$P$<0.05; **$P$<0.01; ***$P$<0.001; ****$P$<0.0001; ns, not significant as compared to the parental cell line (unpaired two-tailed $t$-tests).

protein (Shibata et al., 2010; Ogawa-Goto et al., 2007; Savitz and Meyer, 1990). Thus, the role of RRBP1 in peroxisome biogenesis could be mediated through the ER. To analyze whether in addition to the ER, RRBP1 is also localized to peroxisomes, we performed colocalization analysis using confocal microscopy and corresponding antibodies labeling peroxisomes, ER or mitochondria (Fig. 5A). RRBP1 mainly localized to the ER (overlay of RRBP1 with calnexin) and showed some localization to mitochondria [ATPIF1 (ATP5IF1) colocalization with RRBP1], whereas we observed only weak colocalization of RRBP1 with peroxisomes (catalase with RRBP1) (Fig. 5A,B).

To further confirm that RRBP1 does not localize to peroxisomes, we examined RRBP1 protein in isolated peroxisomes using immunoprecipitation (IP), as described by Ray et al. (2020). We generated clonal HEK293T cell lines stably expressing 3×HA–EGFP–PEX26 at either low or high levels and verified peroxisomal localization of tagged PEX26 by confocal microscopy (Fig. S2A,B). The IP procedure yielded highly purified peroxisomes, confirmed by the absence of ER (calnexin) and cytosolic (GAPDH) contamination (Fig. 5C). Importantly, RRBP1 was not detected in the peroxisomal fraction (Fig. 5C). Thus, our data clearly show that RRBP1 does not localize to peroxisomes, instead its role in peroxisome biogenesis is mediated at the level of ER.

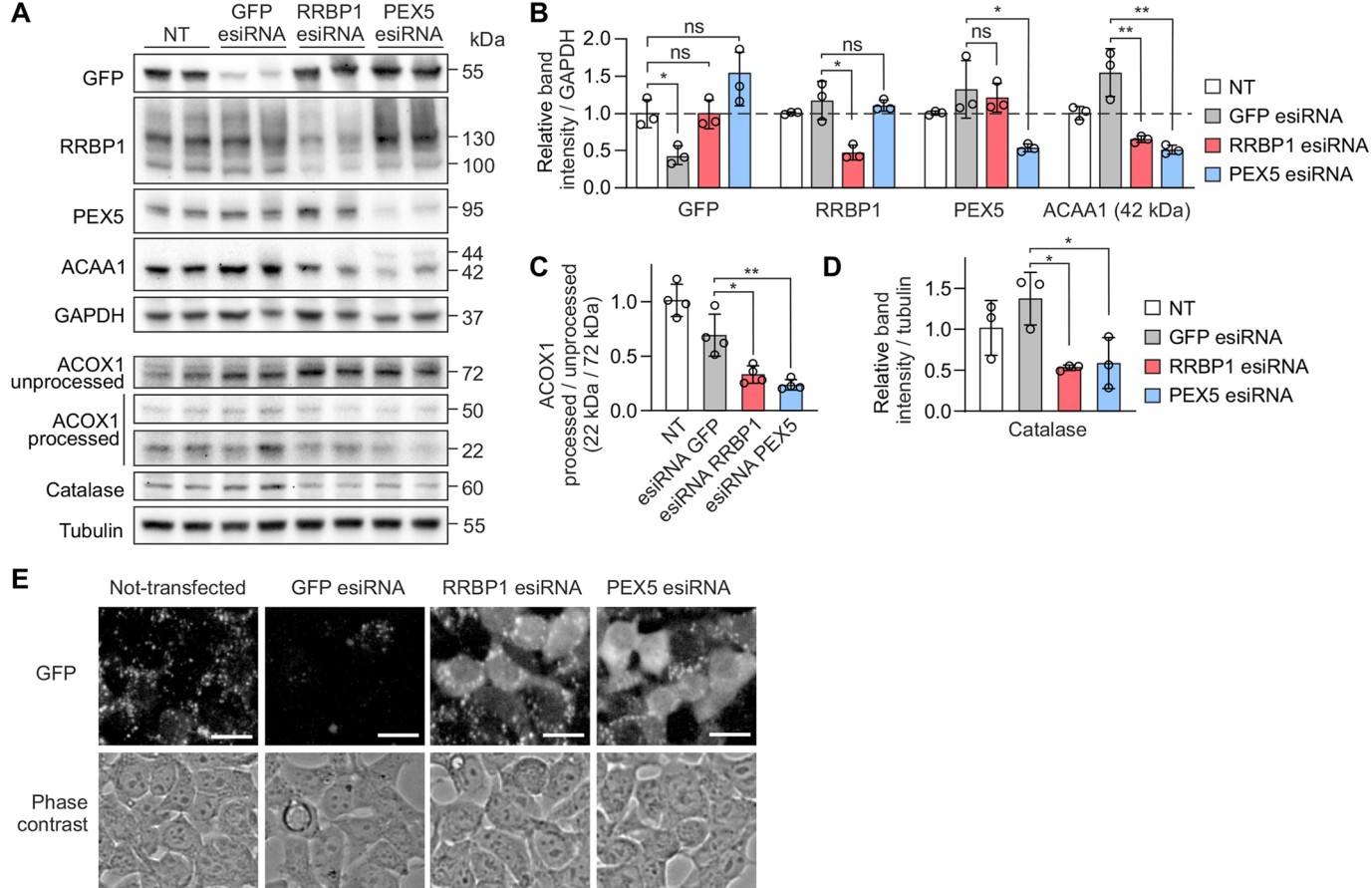

**Fig. 3. RRBP1 silencing causes peroxisomal deficiency in human cells.** (A) Western blot analysis showing the effect of RRBP1 silencing on levels of the indicated proteins in HEK293T cells stably expressing GFP–APEX2–ePTS1. The cells were transfected with corresponding esiRNAs for 72 h. GFP esiRNA was used as a negative control, PEX5 esiRNA was used as a positive control. NT, non-treated cells. (B–D) Quantification of western blot images as presented in A (*n*=3 in B and D, *n*=4 in C). GAPDH or tubulin was used as a loading control. Each western blot analysis was performed at least three times. Values are normalized to the non-treated cells, set at 1. In all graphs data are presented as mean±s.d. *P<0.05; **P<0.01; ns, not significant (unpaired two-tailed *t*-tests). (E) Live-cell fluorescence microscopy images of HEK293T cells stably expressing GFP–APEX2–ePTS1 before (not-transfected) and after transfection with the indicated esiRNAs. The localization of peroxisomal targeted construct GFP–APEX2–ePTS1 was visualized using the GFP channel. GFP esiRNA was used as a negative control, PEX5 esiRNA was used as a positive control. Scale bars: 25 μm. Images are representative of two independent transfection experiments.

## RRBP1 KO causes peroxisomal deficiency independent of ER stress

Since RRBP1 is an ER protein that might influence protein translation and cell proliferation, we tested whether RRBP1 KO cells exhibited altered proliferation. RRBP1 KO did not affect cell proliferation (Fig. S3A), and the protein translation rate measured by puromycin incorporation into a newly synthesized proteins also remained unchanged (Fig. S3B,C). However, we observed a reduction in specific ER proteins, including VAPB, calnexin, IRE1α and BiP (HSPA5), in RRBP1 KO cells compared to the parental cell line (Fig. S3D,E; Fig. 2A,D). This suggests a mild disruption of ER homeostasis. However, the levels of SEC61 and RPL7 were unaltered, indicating that the overall machinery for protein synthesis and translocation, as well as ribosomal assembly, remains unaffected.

## Peroxisome–ER contact sites are not affected in RRBP1-depleted cells

Recent studies suggest that RRBP1 can act as a tether in contact sites between ER and mitochondrial membranes ensuring metabolic communication between two organelles (Hung et al., 2017; Anastasia et al., 2021). To test whether RRBP1 has a role in

maintaining peroxisome–ER contacts we used electron microscopy. We took advantage of APEX2, a well-characterized powerful tool for labeling intracellular structures in electron microscopy studies (Martell et al., 2017; Rae et al., 2021; Sengupta et al., 2019; Lam et al., 2015). Using DAB staining, we labeled peroxisomes in RRBP1 KO HEK293T cells stably expressing GFP–APEX2–ePTS1 and in the cells transfected with esiRNA targeting RRBP1. Parental cells and cells transfected with esiRNA targeting luciferase were used as controls. We observed that neither RRBP1 silencing by esiRNA nor RRBP1 KO affected the number of contact sites between the ER and peroxisomes (Fig. 6A–E). Additionally, electron microscopy analysis of peroxisomal morphology in RRBP1 KO cells revealed no changes in peroxisomal size or shape (Fig. 6F,G).

## The decreased number of peroxisomes in RRBP1 KO cells is independent of pexophagy but is associated with increased proteasomal degradation of peroxisomal membrane proteins

We showed that the steady-state number of peroxisomes is decreased upon RRBP1 KO (Fig. 4A–E). Peroxisomal homeostasis is regulated by the interplay between peroxisome biogenesis and the selective

Journal of Cell Science

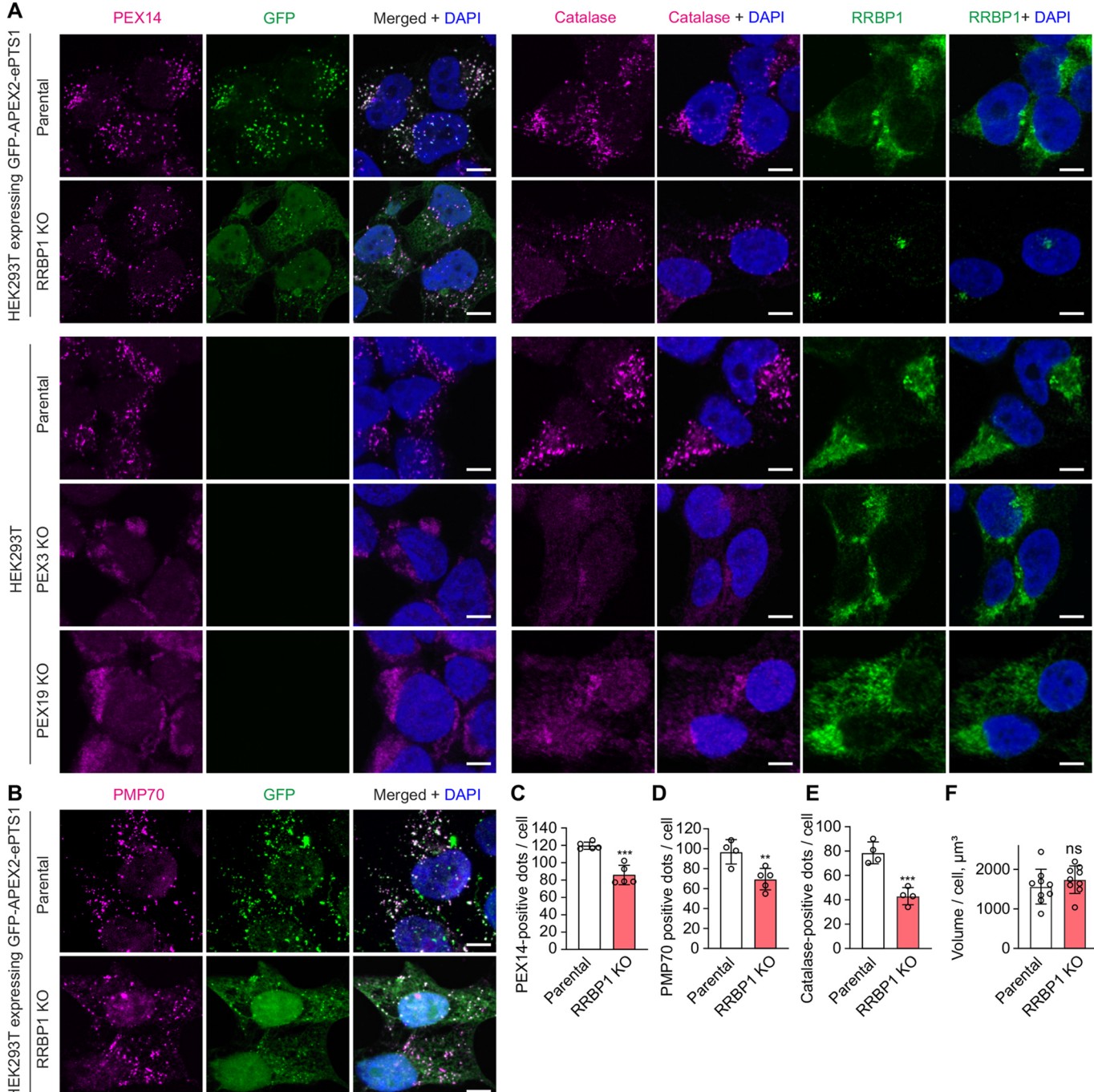

**Fig. 4. The number of peroxisomes is decreased in RRBP1 KO cells.** (A) Confocal microscopy images of RRBP1 KO or parental HEK293T cells stably expressing GFP–APEX2–ePTS1. Parental, PEX3 KO and PEX19 KO cells are shown as controls. Immunostaining was performed using anti-PEX14, anti-GFP, anti-catalase or anti-RRBP1 antibody, as indicated. Scale bars: 5 μm. (B) Confocal microscopy images of RRBP1 KO or parental HEK293T cells stably expressing GFP–APEX2–ePTS1. Immunostaining was performed using anti-PMP70 and anti-GFP antibody. Scale bars: 5 μm. (C–E) Quantification of (C) PEX14-, (D) PMP70- or (E) catalase-positive puncta in RRBP1 KO and parental HEK293T cells stably expressing GFP–APEX2–ePTS1. Z-stacks of confocal microscopy images were processed and reconstructed in a 3D model using Imaris software. Each value represents analysis of ∼100 cells ($n$=4–5). (F) The volume of RRBP1 KO and parental HEK293T cells stably expressing GFP–APEX2–ePTS1 was analyzed using confocal microscopy images. Z-stacks of confocal microscopy images were processed and reconstructed in a 3D model using Imaris software. Immunostaining for each protein was performed in at least three independent experiments. Each value represents analysis of ∼100 cells ($n$=9). PEX14 or catalase signal was used for the analysis. In all graphs data are presented as mean±s.d. **$P$<0.01; ***$P$<0.001; ns, not significant as compared to the parental cell line (unpaired two-tailed $t$-tests).

autophagy of peroxisomes, pexophagy (Demers et al., 2023; Germain and Kim, 2020; Sakai et al., 2006). Therefore, the decreased peroxisome number upon RRBP1 KO could be the result of affected peroxisome biogenesis, where fewer peroxisomes were formed, or induced pexophagy leading to the loss of peroxisomes. To distinguish between these two possibilities, we tested whether pexophagy is induced in RRBP1 KO cells. First, we assessed the steady-state levels of autophagy markers in RRBP1 KO cells and observed a reduction in LC3B (also known as MAP1LC3B) and ATG5 levels compared to the parental cell line (Fig. 7A,B). However, steady-state levels of

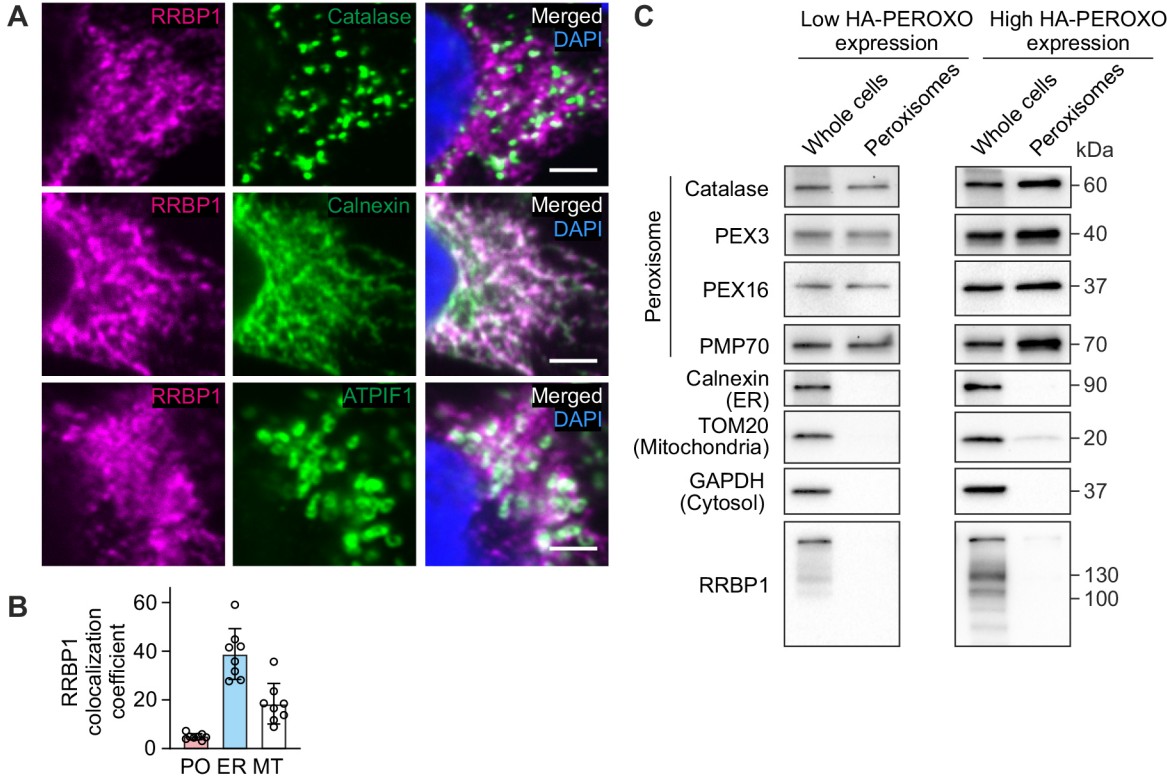

**Fig. 5. RRBP1 does not localize to peroxisomes.** (A) Intracellular localization of RRBP1 was analyzed in HEK293T cells using immunofluorescence confocal microscopy. Catalase was used to label peroxisomes, ATPIF1 was used to visualize mitochondria, calnexin was used to label ER. Scale bars: 3 μm. Immunostaining for each protein was performed in at least three independent experiments. (B) Colocalization analysis of confocal microscopy images as represented in A by Manders' overlap coefficient. The data are presented as mean±s.d. Immunostaining for each protein was performed in at least three independent experiments (*n*=8). MT, mitochondria; PO, peroxisomes. (C) Western blot of the indicated proteins in HEK293T cells stably expressing 3×HA–EGFP–PEX26 (HA–PEROXO) at either low or high levels and immunopurified peroxisomes. Each western blot analysis was performed at least two times.

autophagy markers such as LC3B or ATG5 cannot reliably measure autophagic activity, since autophagy is a dynamic process in which intermediates are continuously generated and degraded. For this reason, we used chloroquine to inhibit pexophagy and assessed whether blocking pexophagy could restore the levels of peroxisomal proteins in RRBP1 KO cells. We confirmed the inhibition of autophagy upon chloroquine treatment by induction of lipidated LC3B (LC3B-II), indicating the accumulation of autophagosomes (Fig. 7A,B). However, chloroquine treatment did not rescue the levels of PEX3 or PMP70 proteins in RRBP1 KO cells (Fig. 7A,B). These results conclusively suggest that pexophagy is not induced in RRBP1 KO cells. Consequently, the reduced number of peroxisomes in cells lacking RRBP1 is likely due to its involvement in peroxisome biogenesis.

To investigate whether peroxisomal proteins undergo proteasomal degradation in the absence of RRBP1, we inhibited the proteasomal degradation pathway using MG132 treatment. Our results revealed that PEX3 and PEX16 were significantly restored in RRBP1 KO cells upon MG132 treatment (Fig. 7C,D), suggesting that their reduction in RRBP1 KO cells is due to proteasomal degradation. Since both PEX3 and PEX16 play crucial roles in initiating peroxisome *de novo* biogenesis via the ER (Aranovich et al., 2014; Kim et al., 2006), their recovery upon proteasomal inhibition indicates that RRBP1 might be involved in the insertion of these proteins into ER membranes. Furthermore, in the parental cell line, MG132 treatment resulted in a significant induction of RRBP1, suggesting that RRBP1 itself has a high turnover rate and might be subject to proteasomal degradation. These findings highlight a potential regulatory mechanism in which RRBP1 helps maintain peroxisomal membrane protein stability, and its absence leads to increased degradation of key peroxisome biogenesis factors.

## DISCUSSION

In this study, we identified and characterized RRBP1, an ER transmembrane protein, as a novel factor involved in peroxisome biogenesis. RRBP1 has been previously implicated in ribosome binding (Savitz and Meyer, 1990), ER–mitochondria tethering (Anastasia et al., 2021; Hung et al., 2017; Cardoen et al., 2024), mitophagy regulation (Killackey et al., 2023) and local axonal translation (Koppers et al., 2024), but its role in peroxisomal function has not been identified.

We showed that cells lacking RRBP1 lose peroxisomes. However, in contrast to cells lacking known essential components of peroxisome biogenesis machinery, like PEX3 and PEX19, RRBP1 KO did not result in complete absence of peroxisomes, indicating that although RRBP1 plays a role in peroxisome biogenesis it is not an essential factor.

Interestingly, in RRBP1 KO cells, as well as in the PEX3 KO cells and PEX19 KO cells completely lacking peroxisomes, we observed induction of PEX5 protein level, a cytosolic peroxin essential for protein delivery into the peroxisomal matrix. These data indicate a complex role for RRBP1 in peroxisomal protein homeostasis. Notably, in cells lacking peroxisomes (PEX3 KO and PEX19 KO cells), the RRBP1 protein level was reduced, suggesting that RRBP1 is responsive to the absence of peroxisomes.

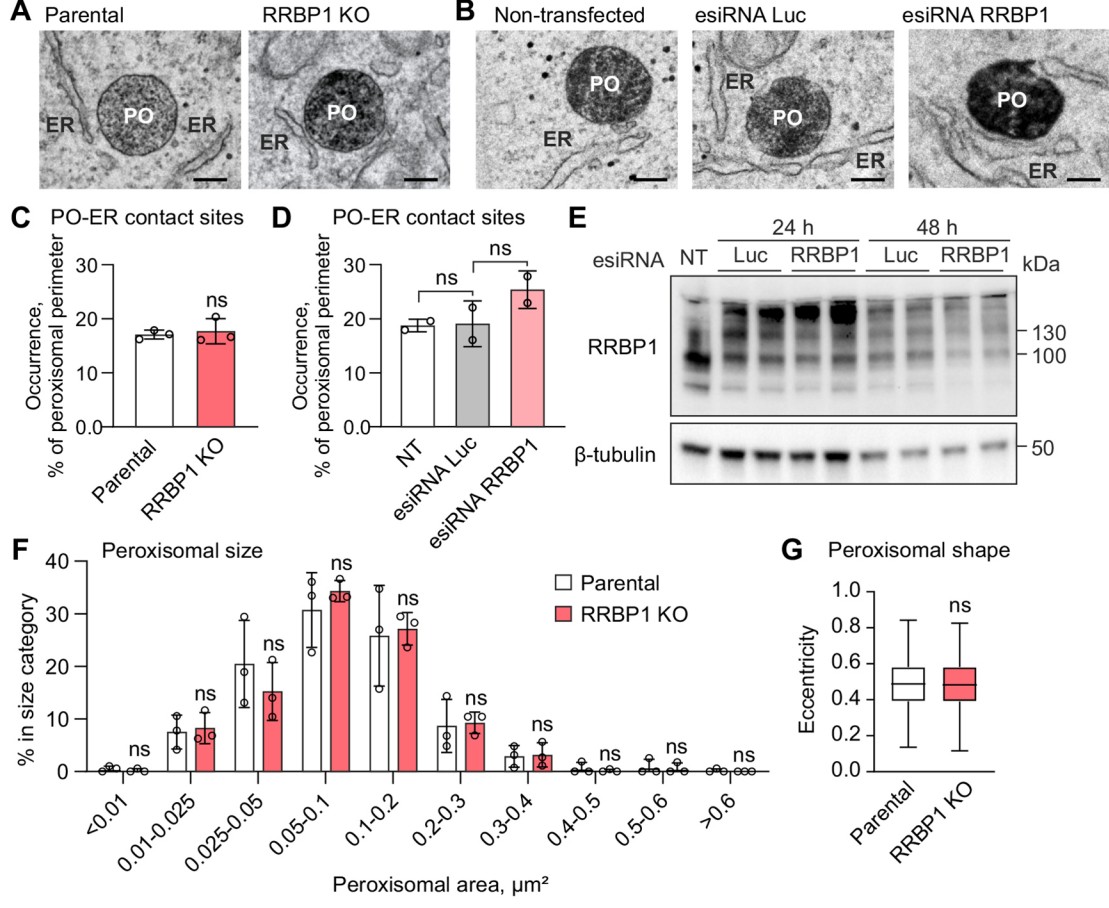

**Fig. 6. Peroxisome–ER contact sites are not affected in RRBP1-depleted cells.** (A) Representative electron microscopy images of RRBP1 KO or parental HEK293T cells stably expressing GFP–APEX2–ePTS1. Peroxisomes were labeled by DAB staining. Scale bars: 200 nm. Experiments were performed in at least two independent replicates for each cell line and condition. (B) Representative electron microscopy images of non-transfected HEK293T cells stably expressing GFP–APEX2–ePTS1 (non-transfected) or the same cell line transfected for 48 h with esiRNAs targeting RRBP1 or luciferase (Luc) as a negative control. Peroxisomes were labeled by DAB staining. Scale bars: 200 nm. (C) Relative occurrence of peroxisome–ER contact sites at 10–100 nm distance in RRBP1 KO or parental HEK293T cells stably expressing GFP–APEX2–ePTS1 analyzed using electron microscopy images. In each experiment for each cell line, 13–18 cells having peroxisomal profiles were selected using systematic random sampling. All the peroxisomes in the selected cells were imaged (*n*=105–175 for each replicate of parental cell line, *n*=84–168 for each replicate of RRBP1 KO cell line, from *n*=3 experiments), and the peroxisome–ER distances were analyzed using Microscopy Image Browser (MIB; Belevich et al., 2016). (D) Relative occurrence of peroxisome–ER contact sites at 10–100 nm distance in non-transfected HEK293T cells stably expressing GFP–APEX2–ePTS1 (NT) or in cells transfected for 48 h with esiRNAs targeting RRBP1 or luciferase (Luc) as a negative control. In each experiment for each cell line, 13–15 cells having peroxisomal profiles were selected using systematic random sampling. All the peroxisomes in the selected cells were imaged, *n*=60–90 for each replicate from *n*=2 experiments, and the peroxisome–ER distances were analyzed using the Microscopy Image Browser (MIB; Belevich et al., 2016). (E) Western blot analysis showing the effect of RRBP1 silencing on levels of the indicated proteins in HEK293T cells stably expressing GFP–APEX2–ePTS1. The cells were transfected with corresponding esiRNAs for 24 or 48 h. Luciferase (Luc) esiRNA was used as a negative control. Each western blot analysis was performed at least two times. (F,G) The distribution of the peroxisomal area (peroxisomal size) analyzed in RRBP1 KO (*n*=437) or parental HEK293T cells (*n*=403) stably expressing GFP–APEX2–ePTS1. Electron microscopy images and Microscopy Image Browser were used for the analysis. Data are from *n*=3 experiments. (G) Eccentricity (peroxisomal shape) was analyzed in RRBP1 KO (*n*=437) or parental HEK293T cells (*n*=403) stably expressing GFP–APEX2–ePTS1 using electron microscopy images and Microscopy Image Browser. Data are from *n*=3 experiments. In C,D and F, data are presented as mean±s.d. ns, not significant as compared to the parental cell line (unpaired two-tailed *t*-tests). In G, line represents the median, box represents the values from the 25th to 75th percentiles and whiskers represent the minimum and maximum values. PO, peroxisome.

To determine whether peroxisome loss in RRBP1 KO cells was due to enhanced peroxisomal degradation, we tested whether blocking pexophagy could restore peroxisomal protein levels. However, inhibition of pexophagy did not rescue peroxisomal protein levels in RRBP1 KO cells. Interestingly, we observed a significant reduction in steady-state LC3B levels in RRBP1 KO cells, which remained lower than in treated parental cells even after pexophagy inhibition. These findings suggest that pexophagy does not contribute to peroxisome loss in RRBP1 KO cells. Instead, the reduced pexophagy rate in RRBP1 KO cells is likely a consequence of the overall decrease in peroxisome abundance.

It is interesting to find that peroxisome biogenesis in human cells requires RRBP1, an ER membrane protein. Notably, RRBP1 KO did not affect cell proliferation or protein translation. Moreover, SEC61 and RPL7 levels remained unchanged in cells lacking RRBP1, suggesting that RRBP1 does not affect translation or translocation machinery.

The classical model of peroxisome biogenesis suggests that peroxisomal membranes originate from the ER in the form of budding pre-peroxisomal vesicles (Lam et al., 2010). However, our APEX2-based proximity labeling suggests that RRBP1 interacts with cargo proteins containing the PTS1 peroxisomal targeting signal,

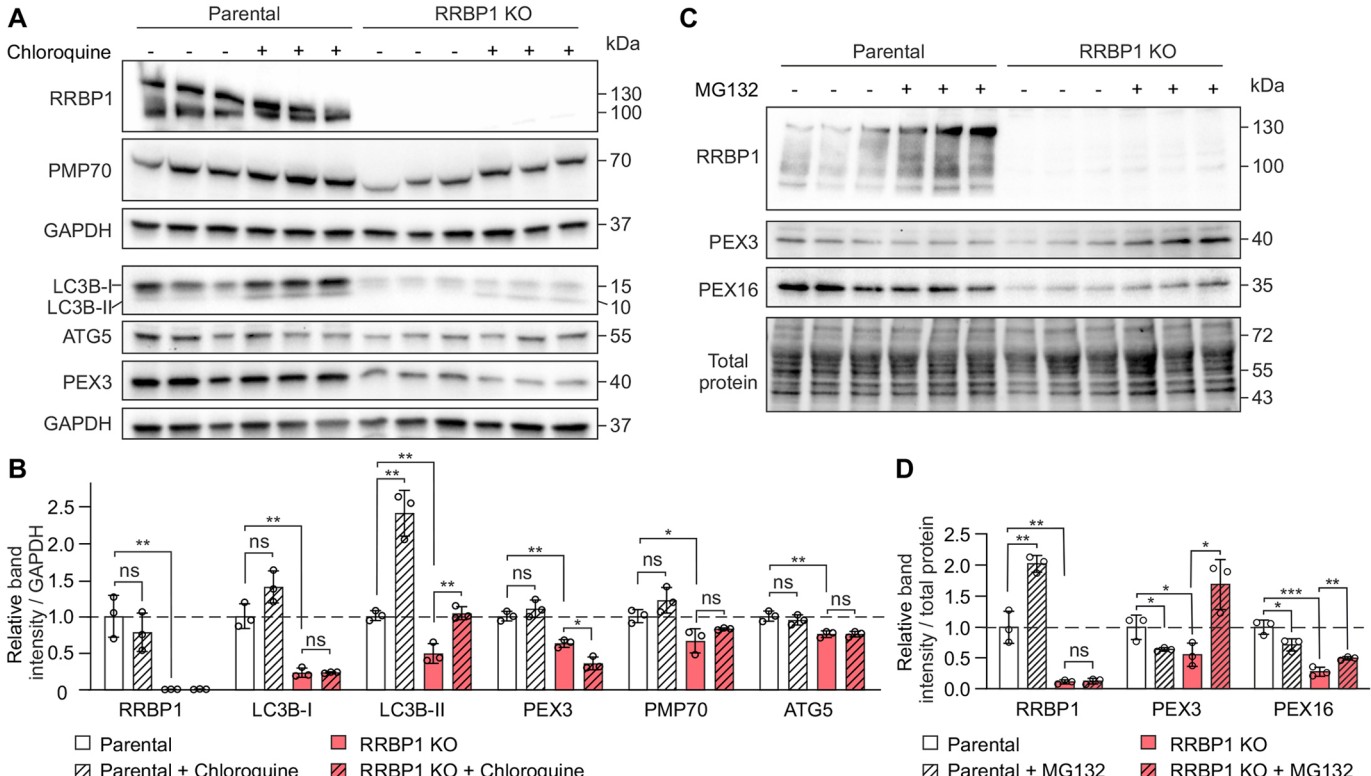

**Fig. 7. The decreased number of peroxisomes in RRBP1 KO cells is independent of pexophagy but is associated with increased proteasomal degradation of peroxisomal membrane proteins.** (A) Western blot analysis of the indicated proteins in RRBP1 KO or parental HEK293T cells stably expressing GFP–APEX2–ePTS1 construct before and after treatment with 100 μM chloroquine for 3 h. GAPDH was used as a loading control. LC3B-I, unlipidated LC3B; LC3B-II, lipidated LC3B. (B) Quantification of western blot images as presented in A ($n$=3). (C) Western blot analysis of the indicated proteins in RRBP1 KO or parental HEK293T cells stably expressing GFP–APEX2–ePTS1 construct before and after treatment with 10 μM MG132 for 4 h. Stain-Free total protein staining was used as a loading control. (D) Quantification of western blot images as presented in C ($n$=3). Each western blot analysis was performed at least two times. Values in B and D are normalized to the parental cell line, set to 1. In all graphs data are presented as mean±s.d. *$P$<0.05; **$P$<0.01; ***$P$<0.001; ns, not significant (unpaired two-tailed $t$-tests).

indicating that RRBP1 has a role in the import of peroxisomal matrix proteins. Importantly, in contrast to peroxisomal membrane proteins, peroxisomal matrix proteins are not detected in the ER (Geuze et al., 2003; Van Der Zand et al., 2010). We also showed that RRBP1 is not localized to peroxisomes, thus making it unlikely that RRBP1 is directly involved in peroxisomal protein import. Instead, our data suggest that RRBP1 might play an indirect role by facilitating the recruitment of other factors to peroxisomal membranes that are required for matrix protein import.

How can an ER protein be involved in peroxisome biogenesis? Recent studies indicate that peroxisomes interact closely with the ER through membrane contact sites, which are crucial for peroxisome dynamics, biogenesis and metabolic signaling (Costello et al., 2017; Hua et al., 2017; Guillén-Samander et al., 2021; Baldwin et al., 2021). However, we did not observe significant changes in the abundance of peroxisome–ER contact sites in RRBP1-depleted cells. However, it is important to note that the DAB staining method used in our study detects only mature peroxisomes, therefore, our data cannot exclude a potential role of RRBP1 in regulating contacts between the ER and pre-peroxisomes.

Our findings indicate a reduction in both peroxisomal membrane and matrix proteins, along with a decreased peroxisome number, suggesting a potential disruption in *de novo* peroxisome biogenesis. We demonstrated that the decreased levels of peroxisomal proteins, such as PEX3 and PEX16, in RRBP1 KO cells result from enhanced proteasomal degradation. Therefore, a potential mechanism by

which RRBP1 contributes to peroxisome biogenesis is through the integration of PEX16 into the ER membrane and its stabilization. PEX16 is known to initiate peroxisome formation by recruiting peroxisomal membrane proteins (Kim et al., 2006; Aranovich et al., 2014; Hua et al., 2015).

Previous studies have shown that RRBP1 promotes the localization of specific mRNAs to the ER surface (Koppers et al., 2024; Cui et al., 2012; Cui and Palazzo, 2014), and a number of mRNAs encoding peroxisomal proteins are enriched on the ER (Haimovich et al., 2016). Additional evidence indicates that targeted mRNA trafficking and local translation contribute to peroxisome biogenesis (Dahan et al., 2022; Zipor et al., 2009). Taken together, these findings support the idea that RRBP1 facilitates the efficient translation of critical peroxisomal proteins, such as PEX16, by directing their mRNAs to the ER. However, the precise mechanism, including whether RRBP1 directly assists in the insertion of peroxisomal proteins or indirectly regulates their recruitment by influencing mRNA localization, remains unclear.

Interestingly, RRBP1 appears to be a metazoan-specific protein: its homologs are not found in yeast. Therefore, despite the conservation of general protein targeting mechanisms in eukaryotic cells, the role of RRBP1 in peroxisome biogenesis could represent a specific pathway of peroxisomal maintenance in higher eukaryotes.

In conclusion, the identification and characterization of RRBP1 as a factor involved in peroxisomal formation advances

our understanding of organelle biogenesis, a complex process involving multiple steps and diverse intracellular cooperation.

## MATERIALS AND METHODS

### Cell culture

HEK293T cells (ATCC, CRL-11268) were cultured (37°C, 5% CO$_2$) in DMEM (Pan-Biotech, P04-03600) supplemented with 10% fetal bovine serum (GIBCO, 10270106), L-glutamine (GIBCO, 25030081) and penicillin-streptomycin (GIBCO, 15140122). The cells were regularly tested for mycoplasma contamination. TransIT-X2 (MirusBio, MIR 6000) or TransIT2020 (Mirus, MIR 5400) were used for cell transfection according to manufacturer's instruction.

### Generation of knockout cell lines

PEX3 KO and PEX19 KO HEK293T cells were generated in our previous study (Somborac et al., 2023) using CRISPR/Cas9 genome editing. Cells were co-transfected with two gRNAs and pSpCas9n(BB)-2A-Puro (PX462) V2.0 (Addgene plasmid 62987; Ran et al., 2013). PEX3 gRNAs were 5′-ACGTGCTTGAAAGGGGGGCAT-3′ and 5′-CACCTCCAAGGA-CCGTGCCC-3′. PEX19 gRNA were 5′-GGGCCCCAGAAGAGAT-CGCC-3′ and 5′-GGGGCCGTGGTGGTAGAAGG-3′. After 24 h, the cells were selected with puromycin and expanded. A T7 assay was used to confirm genome editing. Single-cell clones were obtained and validated using immunoblotting analysis.

To generate RRBP1 KO HEK293T cells expressing EGFP–FLAG–APEX2–ePTS1, an sgRNA transcriptional cassette prepared by PCR and pCAS9-mCherry empty (Addgene plasmid 80975; Schmid-Burgk et al., 2016) were co-transfected into HEK293T cells expressing EGFP–FLAG–APEX2–ePTS1. After 24 h of transfection, mCherry-positive cells were sorted by fluorescence-activated cell sorting (FACS) onto a 96-well plate containing DMEM medium supplemented with 15% fetal bovine serum (GIBCO, 10270106), L-glutamine (GIBCO, 25030081) and penicillin-streptomycin (GIBCO, 15140122), and single-cell clones were generated. The RRBP1 KO cells were validated using immunoblotting analysis. Exon 1 in RRBP1 is non-coding, therefore we used gRNA targeting the first coding exon common to all isoforms of RRBP1, exon 2. The sequence of the sgRNA was 5′-CGAAGGAGGACAGTCACATT-3′.

### Plasmid construction

To generate GFP–APEX2–ePTS1 (pLL3.7-GFP-FLAG-APEX2-ePTS1) we used PCR overlap extension. A FLAG–APEX2 fragment was amplified from pcDNA3 APEX2-NES (Addgene plasmid 49386; Lam et al., 2015), ePTS1 was generated using corresponding primers. The resulting fused construct was inserted into pLL3.7 vector (Addgene plasmid 11795; Rubinson et al., 2003) using EcoRI and NheI restriction sites. To generate GFP–APEX2 (pLL3.7-GFP-FLAG-APEX2), a GFP–FLAG–APEX2 fragment was amplified from pLL3.7-GFP-FLAG-APEX2-ePTS1 plasmid and ligated into pLL3.7 vector. All plasmids were confirmed by Sanger sequencing. Primers used for the cloning are in Table S4.

### Stable cell line generation

HEK293T cells stably expressing GFP–APEX2–ePTS1 or GFP–APEX2 were generated by using standard lentiviral cell line generation protocol. Briefly, wild-type, PEX3 KO or PEX19 KO cells were co-transfected with pMD2.G (Addgene plasmid 12259), psPAX2 (Addgene plasmid 12260) and pLL3.7-GFP-FLAG-APEX2-ePTS1 or pLL3.7-GFP-FLAG-APEX2 using TransIT-2020 (MirusBio, MIR 5400) following the manufacturer's instructions. Single-cell clones were obtained by FACS. The expression of the constructs was confirmed by immunoblotting.

To generate HEK293T cells stably expressing HA–PEROXO (3×HA–EGFP–PEX26), the cells were co-transfected with lentiviral construct pLJC5-3XHA-EGFP-PEX26 (Addgene plasmid 139054; Ray et al., 2020), together with pMD2.G (Addgene plasmid 12259) and psPAX2 (Addgene plasmid 12260) using TransIT-2020 (MirusBio, MIR 5400) following the manufacturer's instructions. Cell populations with low or high GFP expression were subsequently isolated by FACS to establish cell lines.

### APEX2 proximity labeling

APEX2 proximity labeling was performed using a previously described protocol (Hung et al., 2016) with few modifications. HEK293T cells were grown on T75 flasks. The medium was changed and 500 µM biotin–phenol (Iris Biotech, LS-3500) was added to the cell culture medium for 30 min (37°C, 5% CO$_2$). Proximity labeling was induced by adding 10 mM H$_2$O$_2$ (Fisher Scientific, BP2633-500). The reaction was stopped after 1 min, by removing the medium and washing the cells with quencher solution (10 mM sodium ascorbate, 5 mM Trolox and 10 mM sodium azide solution in PBS). Cells were collected in 15 ml quencher solution and pelleted at 3000 $g$ at 4°C for 10 min. Cell pellets were flash frozen in liquid nitrogen. When processing all cell lines was complete, pellets were lysed in RIPA lysis buffer (Cell Signaling Technology, 9806) supplemented with protease inhibitor cocktail (Thermo Scientific, 10320015), 1 mM PMSF, 5 mM Trolox, 10 mM sodium azide and 10 mM sodium ascorbate. Cell lysates were incubated on ice for ~2 min and then centrifuged at 15,000 $g$ at 4°C for 10 min. Streptavidin magnetic beads (Thermo Scientific, 88816) were washed twice with RIPA lysis buffer supplemented with protease inhibitor cocktail (Thermo Scientific, 10320015), 1 mM PMSF, 5 mM Trolox, 10 mM sodium azide and 10 mM sodium ascorbate. Then, clarified cell lysates were added to the beads and incubated overnight with rotation at 4°C. Biotinylated proteins were eluted by boiling the beads in elution buffer [3× protein loading buffer (NEB, P7712S) supplemented with 2 mM biotin and 20 mM dithiothreitol (DTT)] for 10 min. Tubes were briefly centrifuged and, using a magnetic rack, samples were collected, flash frozen in liquid nitrogen and stored at −80°C.

Frozen eluates were thawed on ice and run on 10% separating gel at 200 V for 9 min. Samples were cut from the gel, stained with Coomassie, and washed twice with H$_2$O. Then individual lanes were cut out and stored at −20°C in separate microfuge tubes. Scalpels used to cut the gel were single use, and the samples were loaded with one empty well between.

### Mass spectrometry analysis of proteins

Protein bands were cut out of the SDS-PAGE gel and 'in-gel' digested. Cysteine bonds were reduced with 0.045 M DTT (Sigma-Aldrich, D0632) for 20 min at 37°C and alkylated with 0.1 M iodoacetamide (Fluka, Sigma-Aldrich, 57670) at room temperature in the dark for 30 min. Proteins were digested with 0.75 µg trypsin (Sequencing Grade Modified Trypsin, V5111, Promega) overnight at 37°C. After digestion, peptides were purified with C-18 Micro SpinColumns (The Nest Group Inc. USA) according to the manufacturer's instructions. The dried peptides were reconstituted in 30 µl 0.1% trifluoroacetic acid (TFA) and 1% acetonitrile (ACN) in liquid chromatography–mass spectrometry (LC-MS)-grade water. The mass spectrometry analysis was performed on an Orbitrap Elite and Q Exactive ESI-quadrupole-orbitrap mass spectrometer coupled to an EASY-nLC 1000 nanoflow LC (Thermo Fisher Scientific), using the Xcalibur version 3.1.66.10 (Thermo Scientific) as described previously (Liu et al., 2018). Acquired spectral data files were searched against the human component of UniProtKB/SwissProt database using SEQUEST search engine in the Proteome Discoverer 1.4 (Thermo Scientific). The following parameters were applied: only fully tryptic peptides were allowed with maximum of two missed cleavages, precursor mass tolerance and fragment mass tolerance were set to ±15 ppm and 0.05 Da, respectively. Carbamidomethylation of cysteine was defined as static modification, and oxidation of methionine was set as variable modification. The mass spectrometry data were filtered and presented as peptide spectrum matches.

### esiRNA gene silencing

HEK293T cells expressing EGFP–FLAG–APEX2–ePTS1 were plated in 6-well plate 24 h before transfection. The cells were transfected with 25 nM esiRNA targeting GFP (Merck, EHUEGFP), RRBP1 (Merck, EHU073471) or PEX5 (Merck, EHU090061) using TransIT-X2 (MirusBio, MIR 6000). After 72 h the cells were collected for immunoblotting analysis.

For electron microscopy, HEK293T cells expressing EGFP–FLAG–APEX2–ePTS1 were plated in 6-well plate containing small 13 mm glass coverslips 24 h before transfection. The cells were transfected with 25 nM esiRNA targeting LUC (Merck, EHURLUC) or RRBP1 (Merck, EHU073471), using FuGENE HD Transfection reagent (Promega, E2311). After 72 h, the coverslips were fixed, and samples were prepared for

electron microscopic analysis. The esiRNA cDNA target sequences are in Table S5.

### CRISPR/Cas9 microscopy screen

To perform the CRISPR/Cas9 microscopy screen, for each of the 44 selected candidates and two positive controls (PEX5 and PEX19) two gRNAs were designed, except for CSK21, for which only one target sequence was found with sufficient specificity. gRNAs with U6 overhang at the 5′ end and terminator overhang at the 3′ end were purchased as oligonucleotides (Table S6). U6 was cloned from PX335-dCas9-VP192-PGKpuro using PCR. Obtained U6 was 5′-tailed and amplified using PCR. 3′-tailed terminator was generated using PCR. The gRNA transcriptional cassette was generated by PCR using 5′-tailed U6, 3′-tailed terminator and U6 overhang-gRNA-terminator overhang as described and utilized previously (Balboa et al., 2015; Konovalova et al., 2018, 2023). The sequences of used oligonucleotides are in Table S4.

HEK293T cells expressing EGFP–FLAG–APEX2–ePTS1 were plated in 24-well plates 24 h before the transfection. The cells were co-transfected with a single gRNA and pCAS9-mCherry empty (Addgene plasmid 80975; Schmid-Burgk et al., 2016) using TransIT-2020 (Mirus, MIR 5400). The cells were passaged when needed. Two rounds of transfections were performed for each gRNA, each time the cells were analyzed three times at 7, 10 and 14 days after transfection using an EVOS FLoid Imaging System (Thermo Fisher Scientific). The phenotype of the cells was manually classified based on GFP localization: 'no' (GFP localized exclusively to peroxisomes, no mislocalization), 'mild' (some mislocalized GFP), 'strong' (substantial mislocalized GFP), and 'very strong' (extensive mislocalized GFP). The investigator was not aware of the group allocation during the cell analysis.

### Live-cell fluorescence microscopy

Live-cell microscopy imaging was performed on cells in 6-well or 24-well plates using an EVOS FLoid Imaging System (Thermo Fisher Scientific) equipped with 20× objective or Eclipse TS100 (Nikon) equipped with 40× objective.

### Western blotting

HEK293T cells were lysed in RIPA buffer [Cell Signaling Technology (CST), 9806S] supplemented with protease inhibitor cocktail (Thermo Scientific, 78429). Following 15 min incubation on ice, the samples were centrifuged at 14,000 $g$ for 10 min at 4°C. Protein concentration was measured by bicinchoninic acid (BCA) assay (Thermo Fisher, 23228). Protein lysates were supplemented with a Laemmli sample buffer (60 mM Tris-HCl pH 6.8, 2% SDS, 10% glycerol, 5% β-mercaptoethanol, 0.01% Bromophenol Blue), boiled for 5 min at 95° and resolved on 10% or 7.5% Mini-PROTEAN TGX Precast Gels (Bio-Rad, 4561033, 4561036, 4561023, 4561026). The SDS-PAGE was performed at 100 V for 5 min and then 120 V for 1 h 15 min using SDS running buffer (30 g Tris, 144 g glycine, 10 g SDS to 1 l distilled water). Then the proteins were transferred to a 0.2 μM PVDF membrane (Bio-Rad, 1704150) by Trans-Blot Turbo Transfer System (Bio-Rad) using pre-installed settings (2.5 A for 7 min) for transferring proteins of mixed molecular weight. The membranes were blocked in 5% milk or 5% BSA in TBS with 0.1% Tween 20 (TBST) for 1 h at room temperature. Proteins were incubated with the indicated primary antibodies in TBST containing 1% or 5% BSA overnight at 4°C. The following primary antibodies were used: anti-catalase, 1:1000 (CST, 12980S); anti-RRBP1, 1:1000 (Proteintech, 22015-1-AP); anti-PEX3, 1:1000 (Schmidt et al., 2012); anti-PEX19, 1:1000 (Abcam, 137072); anti-PEX14, 1:1000 (Abcam, 183885); anti-PEX16, 1:1000 (Proteintech, 14816-1-AP); anti-PEX5, 1:1000 (Invitrogen, PA5-58717); anti-calnexin, 1:1000 (Abcam, 22595); anti-PMP70, 1:800 (Sigma, SAB4200181); anti-GFP, 1:1000 (Aves, GFP-1020); anti-TOM20, 1:500 (CST, 42406); anti-ACOX1, 1:1000 (Proteintech, 10957-1-AP); anti-ACAA1, 1:2000 (Sigma-MERCK, HPA007244); anti-acetylated tubulin, 1:1000 (Sigma-MERCK, T7451); anti-GAPDH, 1:10,000 (Sigma-MERCK, G9545); anti-ATG5, 1:2500 (Proteintech, 10181-2-AP); anti-LC3B 1:1000 (Novus biologicals, 600-1384); anti-VAPB, 1:5000 (Proteintech, 14477-1-AP); anti-BiP, 1:1000 (CST, 3177); anti-IRE1α, 1:1000 (CST, 3294); anti-SEC61A, 1:1000 (Abcam, 183046); anti-puromycin antibody, 1:10,000 (Sigma, MABE343); anti-RPL7, 1:1000

(Proteintech, 14583-1-AP); and anti-PXMP2, 1:1000 (Proteintech, 24801-1-AP). Next, the membranes were washed with TBST quickly three times and then three times for 15 min at room temperature on shaker. Then the membranes were incubated with HRP-conjugated secondary antibodies against mouse IgG 1:5000 (CST 7076S), rabbit IgG (CST 7074S) or chicken IgY (Promega G135A) in TBST containing 1% BSA for 1 h at room temperature on a shaker. Then the membranes were washed with TBST quickly three times and then three times for 15 min at room temperature on a shaker. ECL western blotting substrate (Thermo Scientific, 32106) was applied to the membrane, and chemiluminescence was captured by Chemidoc imaging system (Bio-Rad). Quantification of the bands was performed by Image Lab Software (Bio-Rad). Full uncropped blot images are presented in Fig. S4.

### Immunofluorescence

HEK293T cells were cultured on 13 mm coverslips 24 h before fixation. Then the medium was removed, and the cells were washed three times with PBS. The cells were fixed with 4% paraformaldehyde for 10 min at room temperature and washed three times with PBS. Then the cells were permeabilized with 0.2% Triton X-100 in PBS for 15 min at room temperature and washed with PBS. Next DAPI (1:1000 in PBS) was added for 10 min at room temperature. The cells were washed with 0.1% Tween 20 in PBS (PBST) and blocked with 5% BSA in PBST for 2 h at room temperature. The cells were then incubated with corresponding primary antibodies diluted in 5% BSA in PBST overnight at 4°C. The following primary antibodies were used: anti-catalase, 1:400 (CST, 12980S); anti-RRBP1, 1:500 (GeneTex GT5610); anti-calnexin, 1:1000 (Abcam, 22595); anti-PEX14, 1:250 (Abcam, 183885); anti-PMP70, 1:150 (Sigma, SAB4200181); anti-GFP, 1:10,000 (Aves, GFP-1020); and anti-ATPIF, 1:100 (CST 8528). Then the cells were washed with PBST three times for 15 min. The cells were incubated with secondary antibodies diluted 1:1000 in 5% BSA in PBST (anti-mouse IgG Alexa Fluor 555, CST 4409S; anti-mouse IgG Alexa Fluor 647, Invitrogen, A-21235; anti-rabbit IgG Alexa Fluor 555, CST, 4413S; anti-rabbit IgG Alexa Fluor 647, CST, 4414S; anti-chicken IgY Alexa Fluor 488, Invitrogen, A-11039) for 1 h at room temperature and washed with PBST three times for 5 min. Finally, the cells were mounted using antifade mounting medium (Vector Laboratories, H-1700-10). The slides were left to dry overnight at room temperature and then stored at 4°C until analysis. The cells were imaged using an upright SP8 confocal microscope (Leica) with a 63× oil immersion objective.

### Quantitative analysis of immunofluorescence images

Imaris Software was used for the analysis of confocal immunofluorescence images. For colocalization analysis Coloc tool within the Surpass system (version 9.5.1) was used. The images were carefully adjusted manually to distinguish between background and signal. The Manders' overlap coefficient was used to quantify the percentage of overlap. In each sample, 5–8 fields of view were analyzed, with each field capturing ~60–120 cells.

To quantify PMP70-, PEX14- or catalase-positive dots, counting cell number or volume, the confocal microscopy images were processed and reconstructed in a three-dimensional (3D) model using Imaris software.

PMP70-, PEX14- or catalase-positive dots were counted using the 'Dots' function. PMP70, PEX14 or catalase signal was adjusted for the optimal dot detection: X4 diameter (0.5 μm) and model PSF (1.5 μm). The number of PMP70-, PEX14- or catalase-positive dots was normalized to the number of the cells in the specific field of view.

The cell number was counted based on DAPI nuclear staining. Parameters were set with an *XY* diameter of 7 μm to optimize nucleus detection. To analyze the cell volume, PMP70, PEX14 or catalase signal was adjusted to cover the cytosolic area and eliminate any redundant covered regions and debris. Within the Surpass system, a surface was added with default settings, excluding the shortest distance calculations.

### Peroxisome isolation by PEROXO-tag IP

Peroxisomes were isolated as described previously (Ray et al., 2020) with slight modifications. HEK293T cells stably expressing 3×HA–EGFP–PEX26 were cultured in two 10 cm dishes for 24 h. Plates were placed on ice and washed twice with cold PBS, after which cells were scraped into 1 ml cold potassium phosphate-buffered saline (KPBS) and pelleted at 1000 $g$ for

2 min at 4°C. Pellets were resuspended in 1 ml KPBS. For whole-cell lysates, 5 µl of the resuspended pellet was mixed with 50 µl lysis buffer [50 mM Tris-HCl pH 7.5, 150 mM NaCl, 1 mM EDTA and 1% IGEPAL (Sigma-Aldrich, I3021) in ultrapure water] containing protease inhibitors (Thermo Fisher Scientific, 78429). The remaining 995 µl was homogenized on ice using a 2 ml Dounce homogenizer (25 strokes). The homogenate was centrifuged at 1000 $g$ for 2 min at 4°C, and the supernatant was collected. The supernatant was incubated with 200 µl anti-HA magnetic beads (Pierce Anti-HA Magnetic Beads, Thermo Scientific, 88836) on a rotator at 4°C for 3.5 min. Beads were washed 10 times with cold KPBS, with tubes replaced after the first and final washes. Beads were resuspended in 50 µl lysis buffer containing protease inhibitors, incubated on ice for 10 min, and removed using a magnetic stand. The supernatant was centrifuged at 21,000 $g$ for 10 min at 4°C, and the resulting clarified fraction was collected as the purified peroxisome protein sample. Prior to immunoblot analysis, both whole-cell lysates and purified peroxisomes were sonicated in a water bath sonicator for three 30 s cycles. Protein concentrations were determined by BCA assay, and proteins were analyzed by western blotting.

### Pexophagy assay
RRBP1 KO HEK293T cells expressing EGFP–FLAG–APEX2–ePTS1 or the parental cell line were plated on 6-well plates 24 h before the treatment. 100 µM chloroquine was added to the cells cultured in complete DMEM medium. The cells were incubated for 3 h, and after the incubation the cells were collected and used for western blotting analysis.

### Proteosomal inhibition assay
Parental cell line or RRBP1 KO HEK293T cells expressing EGFP–FLAG–APEX2–ePTS1 were plated on 12-well plates 24 h before the treatment. The cells were treated with 10 µM MG132 for 4 h, then the cells were collected and used for western blotting analysis.

### Translation assay
Protein translation rate was analyzed by SUrface SEnsing of Translation (SUnSET) method, which involves puromycin labeling of nascent proteins followed by immunoblotting to detect the labeled proteins (Schmidt et al., 2009). HEK293T cells expressing EGFP–FLAG–APEX2–ePTS1 or RRBP1 KO HEK293T cells expressing EGFP–FLAG–APEX2–ePTS1 were plated in 12-well plates. After 24 h, the cells were treated with 10 µg/ml puromycin and incubated for 20 min in a 37°C, 5% $CO_2$ incubator. Afterwards, the cells were collected and lysed, followed by immunoblot analysis.

### Cell proliferation assay
HEK293T, HEK293T cells expressing EGFP–FLAG–APEX2–ePTS1 or RRBP1 KO HEK293T cells expressing EGFP–FLAG–APEX2–ePTS1 were plated in a 48-well plate at a starting density of 23,000 cells per well. After 24 h, the cells were resuspended and mixed with Trypan Blue stain (1:1). Cell counting was performed using the Bio-Rad TC20 automated cell counter.

### Transmission electron microscopy
HEK293T cells expressing EGFP–FLAG–APEX2–ePTS1 were grown on glass coverslips (thickness #1) and fixed with 2% glutaraldehyde (EM-grade) in 0.1 M sodium cacodylate buffer, pH 7.4, supplemented with 2 mM $CaCl_2$. After fixation for 30 min at room temperature, the samples were washed with sodium cacodylate buffer followed by washing with 50 mM glycine–sodium hydroxide buffer, pH 10.5. For identification of peroxisomes, the cells expressing APEX2 were labeled with 5 mM 3,3′-diaminobenzidine (DAB) and 0.15% $H_2O_2$ in glycine–sodium hydroxide buffer for 20–25 min, on ice, in the dark. The reaction was stopped by washing the samples with glycine–sodium hydroxide buffer, after which the cells were post-fixed with 1% reduced osmium tetroxide in 0.1 M sodium cacodylate buffer for 1 h on ice. After washing, the samples were gradually dehydrated in ethanol and acetone prior to flat embedding (Jokitalo et al., 2001) in epoxy (TAAB 812, Aldermaston, UK). Sections of 60 nm were cut using an Ultracut UC7 ultramicrotome (Leica, Vienna, Austria), post-stained with uranyl acetate and lead citrate, and examined at 100 kV using an HT7800 transmission electron microscope (Hitachi High-Technologies, Tokyo, Japan). Images were acquired using a Rio9 CMOS-camera (AMETEK Gatan Inc., Pleasanton, CA, USA).

### Peroxisome–ER contact site analysis
In each experiment for each cell line, 13–18 cells having peroxisomal profiles were selected using systematic random sampling for the analysis of peroxisomes and peroxisome–ER contact sites. All the peroxisomes in the selected cells were imaged at 8000× nominal magnification with a pixel size of 1.4 nm. Images were contrast normalized and converted to 8-bit using Microscopy Image Browser (MIB; Belevich et al., 2016). For unbiased analysis, the images were anonymized before manually segmenting peroxisomes and ER using MIB. The contact sites were analyzed using the MCcalc plug-in (Lak et al., 2021) in MIB software, with probing distance of 150 nm, smoothing 15 and accepting gaps of 7 pixels. The peroxisomal area (peroxisomal size) and eccentricity (peroxisomal shape) were analyzed using MIB software (Belevich et al., 2016).

### Statistical analysis
All data are presented as mean±s.d. For the statistical analyses with two samples, Student's unpaired two-tailed $t$-tests was performed using Graph Pad Prism Software. All the graphs and heat maps were prepared with Graph Pad Prism Software.

### Acknowledgements
We thank the HiLife Flow Cytometry Unit at the University of Helsinki and Tiina Pessa-Morikawa for FACS service. Proteomics Unit at the Institute of Biotechnology, University of Helsinki, is acknowledged for mass spectrometry analysis of proteins. The Light Microscopy Unit of the Institute of Biotechnology at the University of Helsinki and Marko Crivaro are acknowledged for providing confocal microscopy service. Taina Suntio in the Electron Microscopy Unit (supported by HiLIFE and Biocenter Finland) at the Institute of Biotechnology, University of Helsinki, is acknowledged for EM sample preparation. We are grateful to Maria Vartiainen, Ville Hietakangas, Hien Bui, and Parijat Biswas for valuable discussions and suggestions. We also thank Xiang Le Chua for experimental support and Emilia Kuuluvainen, Leonardo Almeida-Souza, and Laura Hakanpää for their comments on the manuscript.

### Competing interests
The authors declare no competing or financial interests.

### Author contributions
Conceptualization: S.K.; Data curation: S.K.; Formal analysis: K.F., S.K.; Funding acquisition: K.F.; Investigation: K.F., H.V., B.A.A., T.S., A.P., E.J., S.K.; Methodology: K.F., H.V., B.A.A., E.J., S.K.; Project administration: S.K.; Supervision: S.K.; Visualization: K.F., S.K.; Writing – original draft: K.F., S.K.; Writing – review & editing: K.F., V.P., P.K., S.K.

### Funding
This work was supported by the European Research Council Starting Grant (grant 637649 to Cory Dunn), Academy of Finland (Research Council of Finland; grant 331556 to Cory Dunn), Jane ja Aatos Erkon Säätiö (grant 200057 to Cory Dunn), Sigrid Jusèlius Foundation (to Cory Dunn), Finnish Cultural Foundation (to K.F.) and financial support for thesis completion from the University of Helsinki Doctoral School (to K.F.). Open Access funding provided by University of Helsinki. Deposited in PMC for immediate release.

### Data and resource availability
All relevant data and details of resources can be found within the article and its supplementary information.

### First Person
This article has an associated First Person interview with the first author of the paper.

### Peer review history
The peer review history is available online at https://journals.biologists.com/jcs/lookup/doi/10.1242/jcs.264075.reviewer-comments.pdf

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
