## [Peer Review File · Journal of Cell Science]

Ribosome-binding protein 1 maintains peroxisome biogenesis

Kaneez Fatima, Helena Vihinen, Ani Akpinar, Tamara Somborac, Anja Paatero, Eija Jokitalo, Ville Paavilainen, Pekka Katajisto and Svetlana Konovalova
DOI: 10.1242/jcs.264075

Editor: Pedro Carvalho

Review timeline

Original submission:	14 April 2025
Editorial decision:	3 June 2025
First revision received:	3 October 2025
Accepted:	31 October 2025

Original submissionFirst decision letter

MS ID#: jcs.264075

MS TITLE: Ribosome-binding protein 1, RRBP1, maintains peroxisomal biogenesis

AUTHORS: Kaneez Fatima; Helena Vihinen; Bala Ani Akpinar; Anja Paatero; Tamara Somborac; Eija Jokitalo; Ville Paavilainen; Pekka Katajisto; Svetlana Konovalova

ARTICLE TYPE: Research Article

Dear Dr Konovalova,

We have now reached a decision on the above manuscript.

To see the reviewers' reports and a copy of this decision letter, please go to:

As you will see, the reviewers raise a number of substantial criticisms that prevent me from accepting the paper at this stage. They suggest, however, that a revised version might prove acceptable, if you can address their concerns. If you think that you can deal satisfactorily with the criticisms on revision, I would be pleased to see a revised manuscript. We would then return it to the reviewers.

Reviewer 1*Advance summary and potential significance to field*

In the submitted manuscript, Kaneez et al., provide nice evidence suggesting that RRBP1/p180 is required for peroxisome levels. This is a novel finding that is quite surprising and of general interest to the readership of JCS.

Overall this is a well written paper with high quality data. I only have a few minor concerns.

1) The title of the paper suggests that RRBP1 is required for peroxisome biogenesis, but I do not see the data that backs up this assertion. Instead, the data is in line with RRBP1 being required

for the proper insertion of matrix proteins into peroxisomes. Could it be that in the absence of RRB1, de novo peroxisomes are simply defective in the import of matrix proteins? It remains possible that p180 is involved specifically in the formation of new peroxisomes, but in the absence of this data, the authors should tone down their conclusions.

2) RRB1 has been implicated in the localization of many mRNAs to the surface of the endoplasmic reticulum (ER) (PMIDs: 22679391, 24644132, 38815583) and a number of mRNAs encoding peroxisomal proteins are enriched on the ER (see PMID 26367800 for an overview of this data). Thus RRB1 may be required for the efficient translation of key mRNAs into peroxisomal proteins, such as PEX16. Although the authors briefly mention this model, the studies that this model is built on should be cited in the discussion.

3) Minor text corrections:

P3, Line 48. This sentence is missing a capital letter.

P5, Line 32, there is an extra period.

P11, Line 54, "de novo biogenesis via the ER (REF)" should be corrected.

4) The sequences for the esiRNAs and gRNAs should be included in the Methods section.

Reviewer 2

Advance summary and potential significance to field

Peroxisomes are highly metabolic organelles found in most eukaryotic cells. Despite being identified over 70 years ago, many aspects of their biogenesis, and the proteins involved in mediating and regulating this process, remain incompletely understood.

In their manuscript, Fatima et al. employ a proximity labeling system to identify proteins interacting with GFP-APEX2-ePTS1 – a construct designed to be targeted to peroxisomes – in HEK293T cells deficient in mature peroxisomes. Through this proteomic approach, they identify the ribosome binding protein RRB1 as a potential player in peroxisome biology. The manuscript presents various experiments showing that RRB1 influences peroxisome-related phenotypes, leading the authors to propose that RRB1 promotes peroxisome biogenesis and may be involved in the initial insertion of peroxisomal components into the ER membrane.

While the study is thorough and the topic timely, I find that the data do not yet convincingly support the central conclusion that ER-localized RRB1 directly regulates peroxisome biogenesis. The rationale for key experiments is not always clear, and some of the interpretations may be premature given the evidence presented. That said, the work addresses an important question and I believe the manuscript has the potential to make a valuable contribution to the field with additional clarification and experimentation.

Comments for the author

Below are my major and minor comments intended to help the authors improve the manuscript.

Major Comments

Proximity Labeling Strategy

The use of a proximity labeling assay with GFP-APEX2-ePTS1 in peroxisome-deficient cells (PEX3 or PEX19 KO) requires a clearer rationale. Why should proteins found in proximity to a mistargeted PTS1 protein be interpreted as peroxisome biogenesis factors? Could these proteins instead be involved in protein quality control or the translation machinery, rather than specifically in peroxisome biogenesis?

Furthermore, the identification of RRB1 by proximity labeling is not validated by an independent method (e.g., co-IP or another biochemical interaction assay). While it's possible that the interaction is transient and thus detectable only via APEX, a discussion of this limitation would be helpful if other methods did not work.

Interpretation of RRB1 Perturbation Effects

The authors convincingly show that RRB1 KO or KD alters the localization of GFP-APEX2-ePTS1, catalase, and affects maturation of PTS1 and PTS2 proteins. In one experiment, presented in Figure 3, they use PEX14 puncta quantification to suggest that RRB1 affects peroxisome number. Based on the experiment the authors conclude that RRB1 affects peroxisome number.

Alternatively, the results could also imply that RRBP1 affects targeting matrix proteins. Given that PEX14 is part of the matrix protein import machinery, its distribution could be secondarily affected by altered targeting. To substantiate claims about peroxisome abundance, additional markers of PMPs (and not matrix) proteins such as PMP70 (ABCD3) should be used. Note that, if targeting is disrupted, peroxisomes might appear smaller, complicating quantification. At a minimum, analysis with another PMP marker is needed, at least in KO versus control cells. A rescue experiment re-expressing RRBP1 in KO cells would greatly strengthen the specificity of the observed phenotypes. Side note: The authors mention use of an anti-PMP70 antibody in the Methods (Immunofluorescence), but no corresponding data is shown. Please clarify.

RRBP1 Localization

The authors conclude that RRBP1 does not localize to peroxisomes based on immunofluorescence in Figure 5. However, some overlap with catalase is observed, and IF alone may not be sufficient to rule out partial localization. Additional approaches (e.g., cell fractionation or high-resolution microscopy) are needed to substantiate this conclusion. According to BioGRID, RRBP1 has been found to physically interact with peroxisomal proteins such as PXMP2 and SCP2, further warranting a deeper investigation into potential co-localization.

Peroxisome-ER Contact Sites

Following the assumption that RRBP1 is not localized to peroxisomes, and based on previous papers demonstrating the RRBP1 is localized to the ER, the authors examine whether it affects peroxisome-ER contacts using GFP-APEX-ePTS1 and DAB staining. Representative images from control and RRBP1 KO cells showing peroxisomes and their contacts with the ER should be included. Interpretation of the graphs is not possible without understanding the underlying images. Given the observed effects on peroxisome targeting, reliance on the DAB assay (which relies on proper targeting) could bias the analysis toward mature peroxisomes. This potential bias should be acknowledged and addressed in the discussion.

Pexophagy

The authors examine if RRBP1 affects peroxisome number by modulating pexophagy. They show reduced LC3B and ATG5 levels in KO cells, then apply chloroquine to block autophagy and assess peroxisomal protein levels. Please clarify the rationale: if autophagy is already impaired/extremely low in the KO cells (as indicated by lower LC3B and ATG5), would further inhibition with chloroquine be expected to affect pexophagy and rescue peroxisome levels? Additional clarification and context are needed to interpret these results appropriately.

Minor Comments

Introduction

- o Please cite earlier studies indicating mitochondrial contributions to peroxisome biogenesis in mammalian cells.
- o Peroxisomes degrade very long-chain fatty acids, not merely long-chain.
- o Line 48: One of the motivation sentences appears redundant and should be removed.

Results

- o Page 4: Define "ePTS1" upon first mention.
- o Page 6: Clarify why RRBP1 was selected for focused investigation.
- o Page 8: Define "esiRNA."
- o Page 11, Line 54: The citation "(REF)" should be replaced with the appropriate reference.

Figures

- o Figure 1D: The use of 1-3 to denote both the scoring scale and days post-transfection is confusing. Consider revising the labeling.
- o Figure 4B: DAPI staining appears oversaturated. Consider reducing intensity or presenting it in grayscale.
- o Figure S2D: Please briefly comment on the detection of PXMP2 in this experiment and its potential relevance.

Reviewer 3

Advance summary and potential significance to field

In this study, Fatima et. al., report RRBP1 as novel peroxisome biogenesis protein using combination of proximity labeling assay and targeted CRISPR screen. They show that KO or KD of RRBP1 affects processing of peroxisomal proteins, and peroxisome number. They confirm that the observed phenotype is not due ER stress, increased pexophagy, or changes in ER-peroxisome contact but due to proteasomal degradation of peroxisome membrane proteins Pex3 and Pex16. While the observations are interesting as they potentially address how peroxisomal membrane proteins traffic to peroxisomes via insertion in ER membrane, the study is in preliminary stage as it lacks mechanism.

Comments for the author

Major comments-

- Line 22: Explain what is MG132 when used first time in the manuscript.
 - Supplementary Fig 1A: Include Pex5 expression.
 - Scoring method in Fig 1D not clear. If GFP is localized to cytosol there should a change in peroxisome puncta size or number. Therefore, quantification of peroxisome number is required.
 - Check expression of EGFP-APEX2-ePTS1 supporting Figure 1E.
 - Fig 4A could be performed using Ub-RFP-SKL.
 - For Figure 3, it is mentioned that- RRBP1 is involved in maintenance of peroxisome number in human cells. However, peroxisome number is not quantified at this point in the manuscript in Figure 3.
 - Figure 4A images are not consistent with Figure 1E and 3E where peroxisome number do not seem to be lesser than control cells.
 - Figure 4B has RRBP1 staining but no description in the text associated with it.
 - Check Pex14 levels associated with Figure 4B. Also, stain PMP70 to check peroxisome number.
 - Claim in Figure 5 must be supported by subcellular fractionation.
 - Figure 6- EM images must be reported.
- Minor comments-
- Page 11-line 54 REF missing.

First revision

Author response to reviewers' comments

The authors would like to thank the Reviewers for the insightful comments and suggestions, which have helped to improve the manuscript. Detailed responses to each point are provided below.

Reviewers' comments:

Comments from the Reviewers:

Reviewer 1: In the submitted manuscript, Kaneez et al., provide nice evidence suggesting that RRBP1/p180 is required for peroxisome levels. This is a novel finding that is quite

surprising and of general interest to the readership of JCS. Overall this is a well written paper with high quality data. I only have a few minor concerns.

We sincerely thank the Reviewer for the positive assessment of our work and for recognizing the novelty and broader relevance of our findings. We address the minor concerns in detail below.

1) The title of the paper suggests that RRB1 is required for peroxisome biogenesis, but I do not see the data that backs up this assertion. Instead, the data is in line with RRB1 being required for the proper insertion of matrix proteins into peroxisomes. Could it be that in the absence of RRB1, *de novo* peroxisomes are simply defective in the import of matrix proteins? It remains possible that p180 is involved specifically in the formation of new peroxisomes, but in the absence of this data, the authors should tone down their conclusions.

We thank Reviewer for their valuable comment. Peroxisome biogenesis can proceed through two distinct pathways: (i) *de novo* formation and (ii) growth and division of pre-existing peroxisomes. Import of proteins into the peroxisomal matrix is a critical step in both processes (PMID: 26381541; PMID: 28376335). We demonstrated the reduced number of peroxisomes in the cells lacking RRB1, we also showed that this loss is not associated with the enhanced remove of peroxisomes by pexophagy. Based on these findings, we concluded that RRB1 is required for peroxisome formation, whether by *de novo* biogenesis or by division of pre-existing organelles. At the same time, we acknowledge that our data do not distinguish whether RRB1 specifically contributes to *de novo* peroxisome formation. While our proximity labeling assay suggests that RRB1 may participate in peroxisomal matrix protein import, we do not yet have direct evidence to confirm this mechanism. We have revised the text accordingly to present a more measured interpretation of our findings.

2) RRB1 has been implicated in the localization of many mRNAs to the surface of the endoplasmic reticulum (ER) (PMIDs: 22679391, 24644132, 38815583) and a number of mRNAs encoding peroxisomal proteins are enriched on the ER (see PMID 26367800 for an overview of this data). Thus RRB1 may be required for the efficient translation of key mRNAs into peroxisomal proteins, such as PEX16.

Although the authors briefly mention this model, the studies that this model is built on should be cited in the discussion.

We have added the citations in the Discussion.

3) Minor text corrections:

P3, Line 48. This sentence is missing a capital letter.

Corrected

P5. Line 32, there is an extra period.

Corrected

P11. Line 54, "de novo biogenesis via the ER (REF)" should be corrected. Corrected

4) The sequences for the esiRNAs and gRNAs should be included in the Methods section.

All the sequences of gRNAs used in the study and esiRNA cDNA target sequences now in the Supplementary Tables 5 and 6.

Reviewer 2: SUMMARY OF THE ADVANCE MADE IN THIS PAPER AND ITS POTENTIAL SIGNIFICANCE TO THE FIELD

Peroxisomes are highly metabolic organelles found in most eukaryotic cells. Despite being identified over 70 years ago, many aspects of their biogenesis, and the proteins involved in mediating and regulating this process, remain incompletely understood.

In their manuscript, Fatima et al. employ a proximity labeling system to identify proteins interacting with GFP-APEX2-ePTS1 – a construct designed to be targeted to peroxisomes – in HEK293T cells deficient in mature peroxisomes. Through this proteomic approach, they identify the ribosome binding protein RRB1 as a potential player in peroxisome biology. The manuscript presents various experiments showing that RRB1 influences peroxisome-related phenotypes, leading the authors to propose that RRB1 promotes peroxisome biogenesis and may be involved in the initial insertion of peroxisomal components into the ER membrane.

While the study is thorough and the topic timely, I find that the data do not yet convincingly

support the central conclusion that ER-localized RRBP1 directly regulates peroxisome biogenesis. The rationale for key experiments is not always clear, and some of the interpretations may be premature given the evidence presented. That said, the work addresses an important question and I believe the manuscript has the potential to make a valuable contribution to the field with additional clarification and experimentation.

We thank the Reviewer for their thoughtful and constructive comments. In response, we have carried out additional experimental work and revised the manuscript to address each of the concerns raised. Our detailed replies are provided below.

SUGGESTIONS TO AUTHORS

Below are my major and minor comments intended to help the authors improve the manuscript.

Major Comments

Proximity Labeling Strategy

The use of a proximity labeling assay with GFP-APEX2-ePTS1 in peroxisome-deficient cells (PEX3 or PEX19 KO) requires a clearer rationale. Why should proteins found in proximity to a mistargeted PTS1 protein be interpreted as peroxisome biogenesis factors? Could these proteins instead be involved in protein quality control or the translation machinery, rather than specifically in peroxisome biogenesis?

The GFP-APEX2-ePTS1 construct is normally recognized by the peroxisomal protein import machinery and delivered to the peroxisomal matrix in control cells. In contrast, in PEX3 KO or PEX19 KO cells that lack peroxisomes, the construct remains in the cytosol (Supplementary Fig. 1B). Under these conditions, factors involved in recognizing and trafficking ePTS1-containing proteins are expected to accumulate near the GFP-APEX2-ePTS1 construct. Thus, proteins specifically enriched in proximity to GFP-APEX2-ePTS1 in PEX3 KO or PEX19 KO cells are strong candidates for peroxisome biogenesis factors. At the same time, we acknowledge that mistargeted GFP-APEX2-ePTS1 in peroxisome-deficient cells could also be recognized by protein quality control machineries. To address this, we applied a secondary screening step to the candidates identified in the proximity labeling assay, ensuring that only factors with a demonstrable role in peroxisome biogenesis were retained. We have now revised the Results section to clarify this rationale for using GFP-APEX2-ePTS1 in peroxisome-deficient cells.

Furthermore, the identification of RRBP1 by proximity labeling is not validated by an independent method (e.g., co-IP or another biochemical interaction assay). While it's possible that the interaction is transient and thus detectable only via APEX, a discussion of this limitation would be helpful if other methods did not work.

Proximity labeling approaches such as APEX are particularly well suited to capture dynamic and spatially restricted interactions, which may be difficult to detect by conventional biochemical assays. Therefore, for the unbiased systematic identification of peroxisomal biogenesis factors we used APEX2 proximity labeling approach. We acknowledge that while this approach is useful for generating candidate factors, further validation is required to establish their direct role in peroxisome formation. Therefore, we performed the secondary screening using CRISPR based microscopy and then focused our study on the detailed functional analysis of the selected candidate, RRBP1.

Interpretation of RRBP1 Perturbation Effects

The authors convincingly show that RRBP1 KO or KD alters the localization of GFP-APEX2-ePTS1, catalase, and affects maturation of PTS1 and PTS2 proteins. In one experiment, presented in Figure 3, they use PEX14 puncta quantification to suggest that RRBP1 affects peroxisome number. Based on the experiment the authors conclude that RRBP1 affects peroxisome number. Alternatively, the results could also imply that RRBP1 affects targeting matrix proteins. Given that PEX14 is part of the matrix protein import machinery, its distribution could be secondarily affected by altered targeting.

To substantiate claims about peroxisome abundance, additional markers of PMPs (and not matrix) proteins such as PMP70 (ABCD3) should be used.

Note that, if targeting is disrupted, peroxisomes might appear smaller, complicating quantification.

At a minimum, analysis with another PMP marker is needed, at least in KO versus control cells.

A rescue experiment re-expressing RRBP1 in KO cells would greatly strengthen the specificity of the observed phenotypes.

Side note: The authors mention use of an anti-PMP70 antibody in the Methods (Immunofluorescence), but no corresponding data is shown. Please clarify.

We thank the Reviewer for their valuable comments. We performed immunofluorescence analysis of PMP70-positive dots in RRBP1 KO HEK293T cells and the corresponding parental cell line.

Consistent with our PEX14 and catalase analyses, the number of PMP70-positive dots was significantly reduced in RRBP1 KO cells compared with the parental line. These results have been added to Figure 4 B.

We tested whether the RRBP1 KO phenotype could be rescued by reintroducing RRBP1 through transient transfection of RRBP1 KO cells by using pcDNA4 HisMax- V5-GFP-RRBP1 (Addgene plasmid #92150). Successful transfection was confirmed by RRBP1 immunostaining and western blotting. We then assessed peroxisome abundance via PEX14 immunofluorescence and peroxisomal protein levels by western blot for catalase, PEX3, and PMP70. While peroxisome number and protein levels showed a slight increase in the rescue condition compared to KO cells, these changes did not reach statistical significance, likely due to technical limitations such as variable transfection efficiency and dilution of signal in whole-cell lysates. Nonetheless, the data suggest at least partial recovery upon RRBP1 re-expression. Importantly, although we could not fully confirm the specificity of the phenotype through rescue experiment, we validated it using independent depletion approaches. Both CRISPR-Cas9-mediated RRBP1 KO and esiRNA-mediated RRBP1 knockdown produced similar peroxisomal phenotypes, supporting the specificity of the observed effects.

RRBP1 Localization

The authors conclude that RRBP1 does not localize to peroxisomes based on immunofluorescence in Figure 5. However, some overlap with catalase is observed, and IF alone may not be sufficient to rule out partial localization. Additional approaches (e.g., cell fractionation or high-resolution microscopy) are needed to substantiate this conclusion.

According to BioGRID, RRBP1 has been found to physically interact with peroxisomal proteins such as PXMP2 and SCP2, further warranting a deeper investigation into potential co-localization.

We appreciate Reviewer for this comment. Confocal microscopy allows us to visualize the spatial distribution of proteins within the cell, while powerful, it has resolution limitations that might affect the interpretation of colocalization studies (PMID: 21209361). To more rigorously assess whether RRBP1 localizes to peroxisomes, we examined RRBP1 protein in isolated peroxisomes using immunoprecipitation (IP) as described by Ray et al. (Ray et al., 2020). We generated clonal HEK293T cell lines stably expressing 3×HA-EGFP-PEX26 at either low or high levels, and verified peroxisomal targeting of 3×HA-EGFP-PEX26 by confocal microscopy (Supplementary Fig.). The IP procedure yielded highly purified peroxisomes, confirmed by the absence of ER (calnexin) and cytosolic (GAPDH) contamination (Fig. 5C). Importantly, RRBP1 was not detected in the peroxisomal fraction (Fig. 5C). Together, these results demonstrate that RRBP1 does not localize to peroxisomes; instead, its function in peroxisome biogenesis is mediated at the ER.

Although RRBP1 has been reported to physically interact with sterol carrier protein 2 (SCP2) (PMID: 22939629), SCP2 is not exclusively peroxisomal—it is also present in the cytosol and mitochondria. Therefore, this interaction does not demonstrate RRBP1 localization to peroxisomes. Furthermore, PXMP2 was identified as an RRBP1 interactor in a proximity-labeling assay (PMID: 34079125). It is important to note that proximity labeling reflects spatial proximity rather than direct physical interaction. As PXMP2 is a peroxisomal membrane protein, its proximity to RRBP1 could arise from the cytosolic face of peroxisomes, particularly given the well-established contacts between the ER and peroxisomes. Thus, this finding does not contradict our conclusion that RRBP1 resides at the ER and does not localize to peroxisomes.

Peroxisome-ER Contact Sites

Following the assumption that RRBP1 is not localized to peroxisomes, and based on previous papers demonstrating the RRBP1 is localized to the ER, the authors examine whether it affects peroxisome-ER contacts using GFP-APEX-ePTS1 and DAB staining.

Representative images from control and RRBP1 KO cells showing peroxisomes and their contacts with the ER should be included. Interpretation of the graphs is not possible without understanding the underlying images.

Given the observed effects on peroxisome targeting, reliance on the DAB assay (which relies on proper targeting) could bias the analysis toward mature peroxisomes. This potential bias

should be acknowledged and addressed in the discussion.

We thank Reviewer for this important comment. Now we added EM images to Figure 6 and include the limitations of DAB staining to the discussion.

Pexophagy

The authors examine if RRBP1 affects peroxisome number by modulating pexophagy. They show reduced LC3B and ATG5 levels in KO cells, then apply chloroquine to block autophagy and assess peroxisomal protein levels.

Please clarify the rationale: if autophagy is already impaired/extremely low in the KO cells (as indicated by lower LC3B and ATG5), would further inhibition with chloroquine be expected to affect pexophagy and rescue peroxisome levels? Additional clarification and context are needed to interpret these results appropriately.

We appreciate Reviewer for this comment. Since the reduced number of peroxisomes in RRBP1 KO cells could, in principle, result from enhanced pexophagy, we tested whether blocking pexophagy might restore the peroxisomal phenotype. Steady-state levels of autophagy markers such as LC3B or ATG5 cannot reliably measure autophagic activity, since autophagy is a dynamic process in which intermediates are continuously generated and degraded. Nonetheless, changes in their steady-state levels can reflect perturbations in autophagy. For this reason, we used chloroquine to inhibit pexophagy and then assessed peroxisomal protein levels. If increased pexophagy were responsible for peroxisome loss in RRBP1 KO cells, chloroquine treatment would be expected to rescue peroxisomal proteins. Indeed, chloroquine treatment effectively blocked autophagy, as confirmed by LC3B-II accumulation, but it did not restore PEX3 or PMP70 levels in RRBP1 KO cells (Fig. 7A, B). These results demonstrate that pexophagy is not induced in the absence of RRBP1. Instead, the reduced number of peroxisomes is most likely due to the role of RRBP1 in peroxisome biogenesis. We have revised the Results and Discussion sections to clarify this rationale and interpretations.

Minor Comments

Introduction

o Please cite earlier studies indicating mitochondrial contributions to peroxisome biogenesis in mammalian cells.

The citation is added to Introduction.

o Peroxisomes degrade very long-chain fatty acids, not merely long-chain.

Corrected

o Line 48: One of the motivation sentences appears redundant and should be removed.

Corrected

Results

o Page 4: Define "ePTS1" upon first mention.

Corrected

o Page 6: Clarify why RRBP1 was selected for focused investigation.

RRBP1 also known as p180/ribosome receptor, is a transmembrane ER protein. RRBP1 was originally identified as a ribosome receptor for the ER (Savitz and Meyer, 1990) and has also been proposed to regulate local translation in axons (Koppers et al., 2024). Recent studies suggest that RRBP1 plays a role in tethering ER with mitochondria (Anastasia et al., 2021; Cardoen et al., 2024; Hung et al., 2017) and regulates mitophagy (Killackey et al., 2023). However, RRBP1 has never been linked to peroxisomal function before. Therefore, RRBP1 emerged as a particularly compelling candidate due to its ER localization, established role in ribosome binding and membrane protein translocation, processes closely tied to organelle biogenesis. Together with the strong phenotype observed in our screen (Fig. 1D, E), these features provided the rationale for selecting RRBP1 for detailed functional characterization as a potential peroxisome biogenesis factor.

This clarification is now included in the Results.

o Page 8: Define "esiRNA."

The esiRNA is defined.

o Page 11, Line 54: The citation "(REF)" should be replaced with the appropriate reference.

Corrected

Figures

o Figure 1D: The use of 1-3 to denote both the scoring scale and days post- transfection is confusing. Consider revising the labeling.

We appreciate Reviewer for the comment. Indeed, this scoring seemed to be confusing, now we change it in Figure 1D as well as in Supplementary Figure 1D.

o Figure 4B: DAPI staining appears oversaturated. Consider reducing intensity or presenting it in grayscale.

DAPI signal has been adjusted in Figure 4.

o Figure S2D: Please briefly comment on the detection of PXMP2 in this experiment and its potential relevance.

The detection of PXMP2 protein level is now mentioned in the results. Consistent with the loss of other peroxisomal proteins, reduced PXMP2 level indicates decreased peroxisome abundance in RRBP1 KO cells.

Reviewer 3: SUMMARY OF THE ADVANCE MADE IN THIS PAPER AND ITS POTENTIAL SIGNIFICANCE TO THE FIELD

In this study, Fatima et. al., report RRBP1 as novel peroxisome biogenesis protein using combination of proximity labeling assay and targeted CRISPR screen. They show that KO or KD of RRBP1 affects processing of peroxisomal proteins, and peroxisome number. They confirm that the observed phenotype is not due ER stress, increased pexophagy, or changes in ER-peroxisome contact but due to proteasomal degradation of peroxisome membrane proteins Pex3 and Pex16. While the observations are interesting as they potentially address how peroxisomal membrane proteins traffic to peroxisomes via insertion in ER membrane, the study is in preliminary stage as it lacks mechanism.

SUGGESTIONS TO AUTHORS

Major comments-

-Line 22: Explain what is MG132 when used first time in the manuscript.

The explanation of MG132 is now added.

-Supplementary Fig 1A: Include Pex5 expression.

WB analysis of PEX5 protein level is included in the Supplementary Fig 1A.

-Scoring method in Fig 1D not clear. If GFP is localized to cytosol there should a change in peroxisome puncta size or number. Therefore, quantification of peroxisome number is required.

Now we change the scoring in Figure 1D as well as in Supplementary Figure 1D. Mislocalization of the GFP signal in reporter cells expressing the peroxisome- targeted construct GFP-APEX2-ePTS1 indicates altered peroxisomal function. We used this reporter cell line together with gRNAs to screen the effects of 44 candidate genes identified in the proximity labeling assay. This system enabled high- throughput microscopy screening without the need for labor-intensive analysis of peroxisome number and morphology, which would otherwise require cell fixation and high-resolution confocal imaging. Once promising candidates such as RRBP1 were identified in this screen, we proceeded with more detailed functional analyses.

-Check expression of EGFP-APEX2-ePTS1 supporting Figure 1E.

We performed immunoblot analysis of EGFP-APEX2-ePTS1 expression in the cells shown in Figure

1E. Overall expression level of EGFP-APEX2-ePTS1 was not changed upon transfection with gRNAs. The corresponding blots are provided in Supplementary Figure 1. We also mentioned this observation in Results.

- Fig 4A could be performed using Ub-RFP-SKL.

In Figure 4A, we present live-cell fluorescence microscopy images of RRBP1 KO and parental HEK293T cells generated in this study and used throughout the project. These cells express the peroxisome-targeting construct GFP-APEX2-ePTS. While alternative constructs such as Ub-RFP-SKL or mCherry-SKL could also have been employed, we chose GFP-APEX2-ePTS because we had already established HEK293T cells expressing this construct for the systematic identification of peroxisomal biogenesis factors (Fig. 1). We therefore utilized the same cell line background to generate the RRBP1 KO cells.

-For Figure 3, it is mentioned that- RRBP1 is involved in maintenance of peroxisome number in human cells. However, peroxisome number is not quantified at this point in the manuscript in Figure 3.

We thank the reviewer for this helpful comment. Now we corrected this discrepancy.

-Figure 4A images are not consistent with Figure 1E and 3E where peroxisome number do not seem to be lesser than control cells.

We appreciate the reviewer for the comment. Each of the images represents a different cell line or treatment. Figure 1E shows cells transiently transfected with a gRNA targeting RRBP1, 13 days post-transfection. Figure 3A shows cells transiently transfected with esiRNA targeting RRBP1, 72 hours post-transfection. In both cases (Figures 1E and 3A), the cell populations are heterogeneous, with some cells affected by transfection and others remaining unaffected. By contrast, Figure 4A shows an RRBP1 knockout cell line generated by single-cell cloning. This population is relatively homogeneous, as all cells lack RRBP1 protein. Consequently, the reduction in peroxisome number is consistent and clearly observed in this cell line (Figure 4A).

-Figure 4B has RRBP1 staining but no description in the text associated with it. The description is added to the text.

-Check Pex14 levels associated with Figure 4B. Also, stain PMP70 to check peroxisome number.

WB analysis of PEX14 protein levels associated with Figure 4B is added to Figure 2A and B. Also, we performed immunofluorescence analysis of PMP70-positive dots in RRBP1 KO HEK293T cells and the corresponding parental line. Consistent with our PEX14 and catalase analyses, the number of PMP70-positive dots was significantly reduced in RRBP1 KO cells compared with the parental line. These results have been added to Figure 4.

-Claim in Figure 5 must be supported by subcellular fractionation.

To more rigorously assess whether RRBP1 localizes to peroxisomes, we examined RRBP1 protein in isolated peroxisomes using immunoprecipitation (IP) as described by Ray et al. (Ray et al., 2020). We generated clonal HEK293T cell lines stably expressing 3×HA-EGFP-PEX26 at either low or high levels, and verified peroxisomal targeting of 3×HA-EGFP-PEX26 by confocal microscopy (Supplementary Fig.). The IP procedure yielded highly purified peroxisomes, confirmed by the absence of ER (calnexin) and cytosolic (GAPDH) contamination (Fig. 5C). Importantly, RRBP1 was not detected in the peroxisomal fraction (Fig. 5C). Together, these results demonstrate that RRBP1 does not localize to peroxisomes; instead, its function in peroxisome biogenesis is mediated at the ER.

-Figure 6- EM images must be reported.

The represented EM images are included in Figure 6A.

Minor comments-

-Page 11-line 54 REF missing.

Corrected

Second decision letter

MS ID#: jcs.264075R1

MS Title: Ribosome-binding protein 1, RRBP1, maintains peroxisome biogenesis

Authors: Kaneez Fatima; Helena Vihinen; Bala Ani Akpinar; Tamara Somborac; Anja Paatero; Eija Jokitalo; Ville Paavilainen; Pekka Katajisto; Svetlana Konovalova

Article Type: Research Article

Dear Dr Konovalova,

I am happy to tell you that your manuscript has been accepted for publication in Journal of Cell Science, pending standard publication integrity checks and fixing a few very minor textual changes as suggested by reviewer #2.

Reviewer 2

Advance summary and potential significance to field

I very much enjoyed reading the revised version of the manuscript.

The authors carefully addressed all the reviewers' comments.

The addition of several experiments, such as the quantification of peroxisome number using an anti-PMP70 antibody, peroxisome isolation, the inclusion of representative EM images, and various clarifications, remarkably strengthened the manuscript's overall message.

I strongly support publication of this work in JCS, with only minor comments left to the editor's discretion:

- 1.Introduction, p.3 - Since PEX26 does not directly mediate protein translocation to the peroxisomal matrix, I suggest deleting the words "and PEX26."
- 2.Results, p.4 - Please add the reference to the paper in which the term ePTS1 was first used. As far as I know, it was DeLoache et al. 2016 (PMID: 27025684).
- 3.Results, p.7 - The effects on ACAA1 and ACOX1 maturation were clear in the RRBP1 KO cells, but weaker than those observed in the PEX3 or PEX19 KOs. Therefore, I recommend deleting the word "similar" from the sentence:
"Interestingly, a similar significant decrease in the processing of ACOX1 and ACAA1 was observed in the RRBP1 KO cells."
- 4.Figure 4 - The effects shown for GFP-APEX2-ePTS1 differ between panel A (PEX14 staining) and panel B (PMP70 staining), even though the same cells were used and the only difference is the antibody used to visualize peroxisomes. Please check whether the images shown in the two panels are representative.
- 5.Figure 5C - Please clarify what distinguishes the "low" and "high" HA-PEROXO expression conditions.